

# The AERONET Version 3 aerosol retrieval algorithm, associated uncertainties and comparisons to Version 2

Alexander Sinyuk[1,2], Brent N. Holben[2], Thomas F. Eck[3,2], David M. Giles[1,2], Ilya Slutsker[1,2], Sergey Korkin[3,2], Joel S. Schafer[1,2], Alexander Smirnov[1,2], Mikhail Sorokin[1,2], and Alexei Lyapustin[2]

[1]Science Systems and Applications, Inc. (SSAI), Lanham, MD 20706, USA
[2]NASA Goddard Space Flight Center (GSFC), Greenbelt, MD 20771, USA
[3]Universities Space Research Association (USRA), Columbia, MD 21046, USA

*Correspondence to*: Alexander Sinyuk (aliaksandr.sinyuk-1@nasa.gov)

**Abstract**. The Aerosol Robotic Network (AERONET) version 3 (V3) aerosol retrieval algorithm is described, which is based on the version 2 (V2) algorithm with numerous updates. Comparisons of V3 aerosol retrievals to those of V2 are presented, along with a new approach to estimate uncertainties in many of the retrieved aerosol parameters. Changes in V3 aerosol

retrieval algorithm include: 1) a new polarized radiative transfer code (RTC) , which replaced the scalar RTC of V2, 2) detailed characterization of gas absorption by adding $NO_2$ and $H_2O$ to specify total gas absorption in the atmospheric column, specification of vertical profiles of all the atmospheric species, 3) new Bidirectional Reflectance Distribution Function (BRDF) parameters for land sites adopted from the MODIS BRDF/Albedo product, 4) a new version of the extraterrestrial solar flux spectrum, and 5) new temperature correction procedure of both direct sun and sky radiance measurements. The potential effect

of each change in V3 on single scattering albedo (SSA) retrievals was analyzed. The operational almucantar retrievals of V2 versus V3 were compared for four AERONET sites: GSFC, Mezaira, Mongu, and Kanpur. Analysis showed very good agreement in retrieved parameters of the size distributions. Comparisons of SSA retrievals for dust aerosols (Mezaira) showed a good agreement in 440 nm SSA while for longer wavelengths V3 SSAs are systematically higher than those of V2 with the largest mean difference at 675 nm due to cumulative effects of both extraterrestrial solar flux and BRDF changes. For non-

dust aerosols, the largest SSA deviation is at 675 nm due to differences in extraterrestrial solar flux spectrums used in each version. Further, the SSA 675 nm mean differences are very different for weakly (GSFC) and strongly (Mongu) absorbing aerosols which is explained by the lower sensitivity to a bias in aerosol scattering optical depth by less absorbing aerosols. A new hybrid (HYB) sky radiance measurements scan is introduced and discussed. The HYB combines features of scans in two different planes to maximize the range of scattering angles and achieve scan symmetry, thereby allowing for cloud screening

and spatial averaging which is an advantage over the principal plane scan that lacks robust symmetry. We show that due to extended range of scattering angles HYB SSA retrievals for dust aerosols exhibit smaller variability with SZA than those of almucantar (ALM) which allows extending HYB SSA retrievals to solar zenith angles (SZA) less than 50° to as small as 25°. The comparison of SSA retrievals from closely time matched HYB and ALM scans in the 50° to 75° SZA range showed good agreement with the differences below ~0.005. We also present an approach to estimate retrieval uncertainties which utilizes

the variability in retrieved parameters generated by perturbing both measurements and auxiliary input parameters as a proxy



for retrievals uncertainty. The perturbations in measurements and auxiliary inputs are assumed as estimated biases in aerosol optical depth (AOD), radiometric calibration of sky radiances combined with solar spectral irradiance, and surface reflectance. For each set of Level 2 Sun/sky radiometer observations, 27 inputs corresponding to 27 combinations of biases were produced and separately inverted and to generate the following statistics of the inversion results: average, standard deviation, minimum

and maximum values. From these statistics standard deviation (labeled as U27) is used as a proxy for estimated uncertainty and a lookup table (LUT) approach was implemented to reduce the computational time. The U27 climatological LUT was generated from the entire AERONET almucantar (1993-2018) and hybrid (2014-2018) scan database by binning U27s in AOD (440 nm), Angstrom Exponent (AE, 440-870nm), and SSA (440, 675, 870, 1020 nm). Using this LUT approach, the uncertainty estimates U27 for each individual V3 Level 2 retrieval can be obtained by interpolation using the corresponding measured and

inverted combination of AOD, AE, and SSA.

## 1.    Introduction

The optical properties of particles in the earth's atmosphere are measured or retrieved from numerous platforms

including space-based, airborne (or suborbital) and surface-based monitoring instruments. Remote sensing of aerosol optical properties is performed from all three of these instrument deployment platform types while in situ measurements are made from both airborne and surface-based platforms. Measurements from all three of these major categories of platforms of both remotely sensed plus in situ data, in conjunction with atmospheric modeling, will be required to more accurately and completely understand the global extent of variability in aerosol optical properties (Kahn et al., 2004). While satellite (both

polar orbiting and geostationary) observations and retrievals are required for a full spatial understanding of aerosol properties, the uncertainty of these retrievals are at times greater than required for some applications. Additionally, algorithm development for satellite retrieval of aerosol optical properties typically requires a highly accurate set of measurements from which to gauge the retrievals and include some a priori assumptions of particular properties of major aerosol types. The Aerosol Robotic Network (AERONET) (Holben et al, 1998) of globally distributed ground-based instruments has provided the basis for total

column aerosol optical properties that are required for satellite validation purposes and for some algorithms the specification of particular aerosol optical properties that must be assumed in the retrieval algorithms. Satellite validation has been mainly focused on aerosol optical depth (AOD) and accurate AERONET measurements of this parameter have been utilized for numerous satellite instruments and algorithms (Sayer et al., 2019; Sayer et al., 2018; Levy et al., 2013; Levy et al., 2015; Holzer-Popp et al., 2013; Lyapustin et al., 2018; Kahn et al., 2010; Limbacher et al., 2019; Ahn et al., 2014; Choi et al., 2018).

Some satellite algorithms require that aerosol absorption be specified a priori and some utilize AERONET retrievals of imaginary refractive index and/or single scattering albedo (SSA) as the source of these regional values (Remer et al., 2005; Lyapustin et al., 2018). Additionally, some initial aerosol size distributions of basic types are required for some satellite retrievals of AOD and for some algorithms these have been developed or informed by analysis of AERONET retrievals (Remer et al., 2005; Holzer-Popp et al., 2013; Lyapustin et al., 2018). Furthermore, assessment of satellite retrievals of aerosol



absorption by Jethva et al. (2014), as parameterized by SSA, has been performed by comparison to AERONET retrievals of SSA.

In order to assess the impact of aerosol on the earth climate system and compute aerosol radiative forcing, effective total column parameters are often required. AERONET measurements of AOD and retrievals of the size distributions and SSA are representative of the radiatively effective total column extinction weighted values. Kinne (2019a) has utilized the

AERONET database of AOD and retrieved parameters in conjunction with models and satellite data to construct a monthly global aerosol climatology. This aerosol climatology was then applied to compute global aerosol radiative forcing in Kinne (2019b). Additionally, analysis of aerosol black carbon in the global atmosphere and its effects on radiative forcing by Bond et al. (2013) utilized AERONET retrievals of SSA and imaginary refractive index as one of many individual data sets of the spatial distribution of aerosol absorption. A particular strength of the AERONET database in analysis of regional or global

aerosol variability is that the instrumentation is consistent throughout the network with the same calibration standards applied to all data, plus all of the data are processed with the same algorithms for the computation of AOD and the retrievals of all other aerosol parameters.

At each AERONET site the key instrumentation is comprised of an automatic sun and sky scanning ground-based radiometer that performs measurements of direct solar intensity and directional sky radiance distributions. AERONET sites

are located at numerous locations worldwide on all continents and some oceanic islands (~500 sites in 2020). The automatic tracking Sun and sky scanning radiometers (CIMEL Electronique CE-318) take direct Sun measurements with a 1.2° full field of view every ~5 to 15 minutes at 340, 380, 440, 500, 675, 870, 940, 1020 nm and 1640 nm (1640 nm excluded in older instruments). These solar extinction measurements are used to compute total column AOD at each wavelength except for the 940 nm channel, which is used to retrieve total column water vapor (or perceptible water) in centimetres (Schmid et al., 2001;

Smirnov et al., 2004; Giles et al., 2019). The estimated uncertainty in computed AOD, due primarily to calibration uncertainty, is ~0.010-0.021 for field instruments (which is spectrally dependent with the higher errors in the UV; Eck et al., (1999a)). In addition to AOD from direct sun measurements, the CIMEL collects sky radiance measurements in the almucantar geometry (fixed elevation angle equal to solar elevation, and ±180° azimuthal sweeps) at 440, 675, 870, and 1020 nm (nominal wavelengths with 380, 500 and 1640 nm added in newer instruments). Almucantar sky radiance measurements are made at

optical air masses of 4, 3, 2, and 1.7 (75°, 70°, 60°, 54° solar zenith angle respectively) in the morning and afternoon, and once per hour in between. The newest instruments also perform Hybrid scan measurements of directional sky radiances allowing additional retrievals below 50° to 25° SZA, which will be discussed more detail later in Section 4.

The spectral AOD combined with the spectral sky radiances measurements constitute the input datasets to AERONET aerosol retrieval algorithm. The algorithm was developed in Dubovik and King (2000b) and Dubovik et al. (2006) and retrieves

column integrated aerosol size distribution and complex index of refraction. Other aerosol characteristics such as single scattering albedo (SSA), absorption AOD, asymmetry factor, lidar and depolarization ratios are calculated from the retrieved aerosol parameters. The stable performance of the inversion algorithm was illustrated in sensitivity studies performed by (Dubovik et al., 2000a) where the perturbations of the inversion resulting from random errors, possible instrument offsets and



known uncertainties in the atmospheric radiation model were analyzed. Dubovik et al. (2000a) employed retrieval tests using
known size distributions to demonstrate successful retrievals of mode radii and the relative magnitude of modes for various
types of bimodal size distributions such as those dominated by a sub-micron accumulation mode or distributions dominated
by super-micron coarse mode aerosols. Based on this retrieval approach and an eight-year record of AERONET aerosol
measurements, a climatology of the microphysical and optical properties of key aerosol types was developed (Dubovik et al.,
2002). Over the years, AERONET aerosol retrievals have been widely used by aerosol remote sensing community (e. g.
Schuster G. et al., 2005, 2016; Xu F. et al., 2018; Chen Q. et al., 2019; Wang J. et al., 2018; Benkhalifa J. et al., 2017; Lee J.
et al., 2017; Qu Y. et al., 2017; Zhang Y. et al., 2017; Si Y et al., 2017).

Although very few direct comparisons of size distribution between in situ and AERONET retrievals have been
published, several aerosol microphysical and optical parameters have been compared in specific regions: size of the fine mode
aerosols (e. g. Schafer et al., 2019; Reid et al., 2005 (South America, southern Africa, and North America); Clarke et al., 2002
(pollution in the Arabian Sea)), size of larger sub-micron aerosols (e. g. Eck et al., 2010 (stratospheric aerosols); Reid et al.,
2006 (sea salt), 2008a (desert dust); Smirnov et al., 2003 (maritime aerosol); Johnson and Osborne, 2011(Sahel region of West
Africa)). These comparisons showed a reasonable good agreement between AERONET retrievals of size distribution and in
situ measurements. For example, (Reid et al., 2005) showed that volume median diameters of the in situ versus the AERONET
retrievals are often within ~0.01 μm of each other. For aerosol absorption, Schafer et al. (2014) compared SSA derived from
AERONET and in situ aerosol profiles measured by the NASA Langley Aerosol Group Experiment (LARGE) team in the
summer of 2011 during the coincident DRAGON-MD (Distributed Regional Aerosol Gridded Observational Network-
Maryland) and DISCOVER-AQ (Deriving Information on Surface conditions from Column and Vertically Resolved
Observations Relevant to Air Quality) experiments. Schafer et al. (2014) determined that the average SSA difference between
LARGE in situ measurements and AERONET retrieval was ~0.01 with a maximum difference in SSA of 0.023, while all of
the observed differences were within the combined stated uncertainty for the AERONET SSA retrieval and measurement
accuracies (0.03 for AERONET; 0.02 for LARGE).

The AERONET aerosol retrieval algorithm gradually evolved from its original version 1 (V1) to V2 though improving
the modeling capabilities of aerosol and land/atmosphere system while keeping the inversion module of the algorithm intact.
The first major advancement in V2 as compared to V1 was the refinement of surface reflectance model. In V1 the reflectance
was assumed to be isotropic and invariant geographically, with reflectance values of 0.03, 0.06, 0.20, and 0.20 for the 440-,
675-, 870-, and 1020-nm wavelengths, respectively. Bright soil and sand surfaces have much higher spectral reflectance than
the values assumed in V1 retrievals which resulted in large errors in prescribed surface reflectance affecting the accuracy of
retrieved aerosol parameters including single scattering albedo (Sinyuk et al., 2007). In V2, bidirectional reflectance
distribution function (BRDF) models were utilized that allow for dynamic reflectance as a function of solar zenith angle over
land and water (e. g. Eck et al., 2008). Over the ocean, the Cox and Munk (Cox and Munk, 1954; Nakajima and Tanaka, 1983)
model approximates the water BRDF as a function of wind speed and over the land the Li-Ross model (e. g. Lucht and Roujean,
2000) was applied. The land BRDF parameters were adopted from MODIS generic ecosystem type models and mixed by the





ecosystem map of Moody et al. (2005). The second major improvement in V2 was related to the modeling of aerosol microphysics. V1 provided two types of aerosol retrievals for each set of input data: one obtained by modeling aerosol particles

as homogeneous spheres and another using a model of randomly oriented spheroids, with either of them to be selected using external information such as Angstrom Exponent (AE). In V2 the aerosol is modeled as a mixture of spherical and non-spherical particles with the percentage of spherical aerosols in the mixture as a new retrieval parameter (Dubovik et al., 2006). The comparison of V1 and V2 aerosol retrievals (Eck et al., 2008) showed improved accuracy of size distribution and SSA retrievals for both fine and coarse mode dominated aerosols. For example, the V1 artificial bi-modality in the coarse mode size

distribution retrieved for dust aerosols was largely eliminated in V2. Also, in V2 the quality assurance criteria of V1 were revised and extended to help improve the accuracy and quality of the retrieved aerosol parameters (Holben et al., 2006).

The principal reasons for updating the V2 aerosol retrieval algorithm to a newer version were to further improve the modeling capabilities and an extend them to a larger number of applications. For example, both V1 and V2 employed a scalar radiative transfer code (RTC) which restricts applicability of the retrieval algorithm to intensity measurements in the visible

and infrared parts of spectrum. For accurate modeling of atmospheric radiation in the UV or application of the retrieval algorithm to polarization measurements, the scalar RTC needed to be replaced by a vector code. Additional improvements involved more accurate characterization of gaseous absorption. In V1 only the absorption by ozone ($O_3$) was accounted for at 675 nm due to its considerable contribution to the total absorption with optical depth of ~ 0.012. Another atmospheric gas which can potentially affect the SSA retrievals at 440 nm is nitrogen dioxide ($NO_2$). $NO_2$ is primarily emitted from the burning

of fossil fuel. $NO_2$ in the lower atmosphere forms principally from emissions from cars, trucks and buses, power plants, and off-road equipment, and in lower concentrations from biomass burning. For some AERONET sites, the $NO_2$ absorption AOD at 440 nm is comparable or even exceeds 675 nm absorption AOD of $O_3$. For example, at the Beijing site (China) the maximum $NO_2$ absorption AOD in December/January exceeds that of $O_3$ absorption in April (in parentheses): ~0.016 (~0.014). Accurately accounting for water vapor ($H_2O$) absorption at 1020 nm is important due to the fact that optical depth of water

vapor can be as high as ~ 0.01 at sites where the column water vapor is very high, even for moderate and small values of water vapor, its contribution to the total absorption can be substantial due to the small 1020 nm AOD for fine mode dominated aerosols and low 1020 nm absorption by dust. In addition, specifying the vertical extinction/absorption profiles for all the atmospheric species can be beneficial especially at short visible wavelengths and UV.

The assessment of the accuracy of aerosol parameters retrieved by the AERONET algorithm was presented in Dubovik

et al. (2000a). The investigation was a series of sensitivity studies for sets of predefined aerosol models for three main aerosol types: water soluble, biomass burning, and dust with input uncertainties due to random errors, instrumental offsets, and known uncertainties in atmospheric radiation modelling. The accuracy of wavelength dependent aerosol parameters (real and imaginary parts of refractive index and SSA) was analyzed by Dubovik et al. (2000a) for only one wavelength (440nm) and a fixed value of SZA ($60^0$), while applicability of the results to the $50^0$-$70^0$ range of SZA was further assumed. Also, the AOD

dependence of uncertainties were analyzed for only two AOD ranges for water-soluble aerosols (smaller and greater than 0.2) and for one AOD range (greater than 0.5) for biomass burning and dust aerosol types. In this study, we extend the analysis of



Dubovik et al. (2000a) by including all the inversion wavelengths and also analyzing the results to a much wider range of both AOD and SZA.

175        This paper describes the changes and additions to the V3 aerosol retrieval algorithm as compared to that of V2, along with a new approach for estimating the uncertainties of retrieved aerosol parameters. The paper is organized as follows: Section 2 describes changes and additions to V3 aerosol retrieval algorithm. In Section 3, we present analysis of the potential effects of each change and addition on the retrieved aerosol parameters, plus comparison of almucantar scan retrievals of aerosol properties from V2 to those of V3. A new measurement protocol (called the Hybrid scan) which increases the SZA range over that of almucantar for small SZAs and allows a robust spectral radiance symmetry check to be performed is discussed in

Section 4. Section 5 presents a new method to estimate the uncertainties of the retrieved aerosol parameters. The final section presents summary and conclusions.

## 2. **Version 3 retrieval algorithm description.**

       Scalar radiative transfer theory can be used to calculate radiances for an aerosol laden atmosphere in the visible part

of electromagnetic spectrum (e. g. Kattawar et al. 1976). However, at shorter wavelengths due to an increasing contribution of molecular scattering, neglecting the effects of polarization results in errors that may be too large for some practical applications (Mishchenko et al. 1994). In Version 3 (V3) of the AERONET aerosol retrieval algorithm a new polarized radiative transfer code (RTC) SORD (Korkin et al., 2017) replaced the scalar discrete ordinate RTC of Nakajima and Tanaka (1986) utilized in Version 2 (V2). SORD allows accurate simulations of atmospheric radiation in a wide spectral range using the method of

successive orders of scattering. The code was extensively tested using 52 benchmark scenarios providing accuracy of ~0.1% for both intensity and polarization calculations. Unlike V2 RTC whose speed was boosted by employing the truncation approximation of Nakajima and Tanaka (1988), no such approximation is used by SORD due to potential problems it can cause in the forward scattering direction (e. g. Korkin et al., 2012). High performance multi-processor computing is utilized to overcome the significant increase in computational resources required with application of SORD in AERONET retrievals.

195        With SORD enabling accurate modeling of atmospheric radiation at short wavelengths the retrieval of aerosol absorption in UV is possible by adding 380 nm channel to standard AERONET spectral range (440 – 1020 nm). The aerosol absorption in UV became the subject of substantial interest in recent years due to the strong absorption of UV radiation by fine mode aerosol (e. g. Corr et. al, 2009). For example, a strong increase of smoke aerosol absorption in UV wavelengths was observed by Mok et al. (2016) and explained by the presence of brown carbon in aerosol composition. Also, dust aerosols

exhibit significant absorption with decreasing wavelength in the visible and UV (e. g. Sokolik and Toon, 1999; Sinyuk, et al., 2003). AERONET can contribute to better understanding of aerosol absorption in UV by providing absorption estimates at numerous locations with varying aerosol types.

       Employing the SORD polarized RTC provides the potential of adding polarization measurements (e.g., CIMEL CE318-DP sun-photometer) as input to the inversion. Recent publications show that combining intensity and polarization

measurements in one dataset improves the accuracy of retrievals of the real part of refractive index of fine mode aerosols and



the shape of aerosol particles but has a little effect on aerosol absorption and size (Li, et al., 2009; Fedarenka, et al., 2016). As of now (2019) AERONET does not widely acquire polarization measurements primarily due to the long time interval of measurement acquisition (thereby creating large temporal gaps in the AOD time series) and the lack of clear understanding regarding the optimal set of spectral channel polarization measurements to be used. Additional research is required to determine
the optimal number of the observational angles and spectral channels needed to reduce the acquisition time of these measurements.

The AERONET V3 aerosol inversion algorithm is the same as the one employed in V2, and is described in Dubovik and King (2000b) and Dubovik et al. (2006). However, V3 incorporates a detailed characterization of gas absorption, which was neglected in V2 and previous versions. In addition to ozone ($O_3$) absorption, V3 also accounts for that of nitrogen dioxide
($NO_2$), and water vapor ($H_2O$) to specify total gas absorption in the atmospheric column. Ozone optical depth is determined utilizing the total column Total Ozone Mapping Spectrometer (TOMS) monthly average climatology (1978–2004) of $O_3$ concentration at 1°. by 1.25° spatial resolution, the $O_3$ optical air mass using $O_3$ scale height adjustment by latitude (Komhyr et al., 1989), and the $O_3$ absorption coefficient of Burrows et al. (1999). Similarly, the nitrogen dioxide optical depth utilized as input to the V3 retrieval is from the total column Ozone Monitoring Instrument (OMI) monthly average climatology (2004–
2013) of $NO_2$ concentration at 0.25° by 0.25° spatial resolution and the $NO_2$ absorption coefficient of Burrows et al. (1998). Total column water vapor amount is determined from the AERONET CIMEL instrument 940 nm channel retrievals (Schmid et al., 1998).

In V2 the atmosphere was modelled as a plane parallel homogeneous layer. While for the almucantar (ALM) observation geometry this is a reasonable assumption (e.g. Dubovik and King, 2000b; Torres et al., 2014), for other geometries
the sensitivity to vertical structure of aerosol and gases in atmosphere can be important, especially at shorter wavelengths with relatively large Rayleigh scattering. In V3 vertical profiles of all input atmospheric species are specified. Daily spectral aerosol extinction profiles are obtained from MERRA-2 (Gelaro et al., 2017) global assimilation model simulations, which utilize CALIOP (e. g. Winker et al., 2007) aerosol vertical profile measurements. The vertical absorption profiles for $O_3$ and $H_2O$ are specified using data of MERRA-2, and NCEP/NCAR Reanalysis data respectively, which are acquired from the NOAA
National Weather Service NOMADS NCEP server (e. g. Saha et al., 2010). The data to characterize typical tropospheric and stratospheric $NO_2$ profiles were obtained from Vlemmix et al. (2015) and Kerzenmacher et al. (2008) respectively. The aerosol extinction, $O_3$, $NO_2$, and water vapor profiles are reported at 73, 73, 20, and 9 atmospheric altitudes respectively. In SORD, they are rescaled to the same grid used for optical depth integration (Korkin et al., 2016).

The surface reflection in SORD is modeled by two Bidirectional Reflection Functions (BRDF): the Cox-Munk model
over water by Nakajima and Tanaka (1983) and the Ross-Li model (e. g. Lucht and Roujean, 2000) over land. BRDF models are important for computing the solar zenith angle dependent spectral reflectance throughout the day. For mixed water/land surfaces the two BRDF models are mixed using the percentage of land and water within a 5 km radius circle centered at each AERONET site. The land/water percentage is determined using MODIS/Terra Land Water Mask. Cox-Munk calculations use near surface wind speed from NCEP/NCAR Reanalysis data. The procedure of assigning parameters to Ross-Li model in V3





is rather different than before. In V2, the land BRDF parameters for each site were determined using the maps of ecosystems types combined with generic BRDF parameters (courtesy of Feng Gao) assigned to each ecosystem type, and gap filled MODIS black sky albedo product (Moody et al., 2005). For each AERONET site the BRDF parameters were averaged over all the ecosystem types within a 5 km radius circle and the resulting BRDF was normalized by MODIS black sky albedo.  Since new BRDF products recently became available, V3 BRDF parameters for land sites are adopted from the MODIS BRDF/Albedo

CMG Gap-Filled Snow-Free Product MCD43GF V005 (see link at the end of publications list). Both V2 and V3 BRDF land parameters were aggregated into a climatology to apply to all years prior to and after the BRDF data availability interval of 2002-2004 for V2 and 2003-2013 for V3.  Near-real-time Ice and Snow Extend (NISE) from the National Snow and Ice Data Center by Nolin et al. (1998) and MODIS snow cover map by Hall et al. (2002) are used to account for the BRDF of snow and/or ice.

In order to improve the overall quality of observations the V3 employs a temperature correction of both AOD and sky measurements applied to account for the effect of high range of instrument sensor head temperature in various environments. The correction procedure and examples of its application are presented in detail in Giles et al. (2019). Note that in V2 only the solar (not sky) 1020 nm channel was corrected for temperature (using a constant nominal value), while in V3 all non-UV channels are corrected for temperature variation, typically using characterization of the temperature sensitivity of

each individual instrument. Also in V3 a newer version of the extraterrestrial solar flux spectrum (Coddington et al.,2016) replaced the older version (ASTM, International, 2007).  The inversion procedure, aerosol parametrization, and the set of retrieved aerosol parameters are the same as those of V2 and described in details in Dubovik and King (2000b) and Dubovik et al. (2004, 2006).

**3. Comparison of AERONET V3 to V2 aerosol retrievals.**

The described changes and additions to the V3 aerosol retrieval algorithm, RTC code and data inputs influence the aerosol retrieval results. This section analyzes the effect these changes have on the V3 inversions by first examining the potential effect of each particular change and then comparing aerosol retrievals from V2 and V3.

**3.1 The effect of changes in input BRDF.**

Figure 1 shows the difference in the black sky surface albedo calculated with Ross-Li BRDF function using parameters of V2 and V3. The calculations are done for four AERONET sites with different surface types: GSFC (USA, MD) - urban/forest, Mezaira (United Arab Emirates) - desert, Mongu (Zambia) – treed savanna, and Kanpur (India) – semiarid/urban with strong seasonality. Figure 1 shows reasonable good agreement among average values of surface albedo

for non-desert sites with maximum difference of ~0.025. For desert, the agreement is good at 440 nm but decreases with increasing of wavelength reaching maximum of ~0.085 at 1020 nm. This discrepancy can be explained by the difference in the angular shape of BRDF functions in V2 and V3 as illustrated in Fig.2a, which presents the difference in surface albedo





as a function of solar zenith angle (SZA) for Mezaira. Figure 2a shows that the surface albedo difference at 440 nm does not exhibit any substantial SZA dependence and therefore can be explained by the difference in the reflectance magnitude. For longer wavelengths, the SZA dependence of surface albedo difference is much stronger resulting in maximum variability from -0.05 to -0.16 at 1020 nm and can be explained by the BRDF of V2 being more anisotropic than that of V3. For non-desert sites the difference in BRDF angular shape is less significant as illustrated by Fig. 2b for the GSFC site, showing albedo differences within ±0.05 which is the reported uncertainty of the MODIS BRDF product (Wang et al., 2018). The large difference in surface albedo for desert sites has the potential for influencing retrievals of single scattering albedo (SSA) at longer wavelengths. Figure 3 illustrates the sensitivity of normalized sky radiances (NSR), which are the input to the inversion code, to the changes in BRDF parameters. The calculations are made for two values of SZA (50° and 75°) using dust aerosol parameters retrieved at the Mezaira site when AOD at 440nm is 0.65 and Angstrom Exponent (440-870nm) is 0.25. As is seen from Fig.3, the BRDF effect on NSR at 440 nm is moderate and decreasing with increasing of SZA (~2.0% to ~ 0.9%). For longer wavelengths the relative difference of NSR is larger reaching −8.8% and −16% at 1020 nm for SZA 50 and 75 degrees respectively.

### 3.2 Effects of changes in Extraterrestrial solar flux and temperature correction.

Figure 4 shows the relative difference between the extraterrestrial solar fluxes of V2 and V3 for the four standard sky wavelengths of the AERONET retrievals. The agreement is reasonable (differences of ~1% or less) at most of the wavelengths except 675 nm where the difference is anomalously large (~3%). Figure 5 shows the combined effects of both changes in solar flux and application of the temperature correction of sky radiances on NSR for two instruments, No 721 and 1026 which were deployed at the Mezaira site in 2013-2014 and 2017-2018 respectively. Each point corresponds to observations taken at different times and also temperatures of the sun photometer sensor head. In addition to temperature, part of the difference is due to solar flux changes and accounting for $NO_2$ and water vapor absorption in V3. For example, at 675 nm the ~ 3% bias is mostly due to differences in solar flux. The change in temperature dependence of the difference is due to the improved temperature correction in V3 (Giles et al., 2019). As is seen from the Fig. 5a, the largest temperature correction effect is for 1020 nm which the most temperature sensitive channel (Holben et al., 1998). At the same time for instrument 1026, the 1020 nm NSR difference is almost temperature neutral. The reason is that instrument 1026 is a latest version –T instrument equipped with new model of filters (Iridian) which is much less temperature sensitive than older filter models (Barr, Spetragon, POC) utilized in older versions of Cimel instruments. Subtraction of the solar flux offset (V3 versus V2) in the case of 1026 instrument gives a maximum NSR difference of ~ 1.3% at 440 nm and less than 1% at 1020 nm. For instruments of older versions, the effect of temperature correction is expected to be larger than for those of Version-T, however, the specific numbers will vary from instrument to instrument.

### 3.3 Joint effects of temperature correction, solar flux and BRDF on retrievals of SSA, case study.



The joint effect of above factors on aerosol retrievals is illustrated on a case study of diurnal variability of dust SSA retrieved at the Hamim AERONET site on August 25 2004 during UAE field campaign (e. g. Reid et al., 2008b). This site is located in a desert region in the United Arab Emirates and is in proximity to several dust aerosol sources. This particular day was selected due to a combination of very stable AOD level (0.486±0.016) and very stable 440-870 nm Angstrom Exponent 310  (0.22 ±0.01) which makes an assumption of the stability of aerosol properties throughout the day reasonable. Also a large temperature gradient (29.1 - 47.8 C), especially in the morning, makes an analysis of the diurnal variability of temperature correction possible. Figure 6a shows the dependence of the difference in NSR on solar zenith angle where negative values correspond to the morning and the positive ones to the afternoon. On Fig. 6b the effect of solar flux is subtracted so the variability is due to difference in temperature correction and BRDF only. Figure6a shows the total range of the variability of 315  NSR difference is within ~ ±4% for 675 and 1020 nm. Comparison of Figures 6a and 6b shows that the 675 nm channel is primarily affected by changes in solar flux while the difference at 1020 nm is mostly due to temperature correction. The diurnal variability of NSR difference at 1020 nm shows clear dependence on SZA, reaching a maximum absolute value in the early afternoon when the instrument sensor head temperature is maximal. The NSR difference at 440 nm and 870 nm are the least affected by solar flux values due to their similarity in both graphs of Figure 6. In addition, comparison of Fig. 6a and Fig. 6b 320  shows that the difference due to the temperature correction at 440 nm is partially compensated by the change in solar flux while at 675 and 1020 nm the both factors work in the same direction.

In analysis of the effect of the changes implemented in the V3 aerosol retrieval algorithm on SSA retrievals from almucantar scans the following approach was used. It consists of generating three different input files for each set of measurements. The NSR and BRDF data of the first input file (1st) are identical to those of V2. The second input file (2nd) 325  differs from the first one by replacing V2 NSR with those of V3 and in the third input file (3rd) both NSR and BRDF parameters are replaced, utilizing only V3 inputs.  These input files are then combined with SORD in different modes, scalar or vector, which allows for analysis of the effects of accounting for polarization on retrieval results. All the combinations are inverted and retrieval results are compared to each other allowing for analysis of sequential changes in retrieved SSA values as individual changes in input data take place. Fig.7a compares the diurnal variability of SSA retrieved by the V2 retrieval 330  algorithm and that of V3 with the 1st file as an input and SORD in the scalar mode.  The comparison shows a good agreement between two sets of retrievals with maximum difference of ~0.0074 at SZA $19.5^0$.  The diurnal variability of SSA retrieved by version three V3 is also shown.

A distinct feature of Fig.7a is a sharp decrease in the retrieved SSA for SZA smaller than $50^0$. This partially can be explained by the decrease in the range of the scattering angles as SZA decreases in which case the solution becomes less stable 335  with respect to both random noise and biases. However, the systematic decrease in retrieved SSA with SZA implies presence of a bias which itself is SZA dependent. The recent analysis of Sinyuk et al. (2015) showed that this bias could be related to the accuracy of the model of randomly oriented spheroids at small scattering angles. This model was originally proposed by Mishchenko et al., (1997) for modeling of light scattering by non-spherical dust aerosols, was further developed in Dubovik et al. (2002, 2006) and is currently employed by AERONET aerosol retrieval algorithm. The model was proved to be successful





in interpreting the results of laboratory measurements of the light scattered by non-spherical aerosols in backscattering direction (Dubovik et al, 2006). However, careful inspection of the fitting of the laboratory measurements by the model shows that it underestimates measured intensity at small scattering angles in the range centered at $\sim 7^0$ (e. g. Lenoble et al., 2013, p. 49, Fig. 2.18, left upper panel). The influence of this bias on the accuracy of SSA retrievals depends on SZA through the range of scattering angles with the maximum scattering angle for almucantar geometry being double of that of SZA. For large SZA

the effect of the bias is counterbalanced by the fit at the large scattering angles where the spheroidal model is accurate. In these cases, the bias is not or just partially fitted by retrieval code and has little effect on the accuracy of SSA retrievals. However, with decreasing of the range of the scattering angles the contribution of the bias to the fitting of NSR increases and with no measurements constraining the solution at larger scattering angles it pushes the retrieved SSA lower. This reasoning is illustrated by the Fig. 7b which shows the relative difference between measured and fitted angular NSR as a function of

scattering angle for the three values of SZA. In particular, all curves exhibit a peak in the vicinity of $\sim 7^0$ where fit underestimates the measurements. The magnitude of the peak is decreasing with decreasing of SZA corresponding to more accurate fit of the bias. Also, a decrease in fitting accuracy for the smallest SZA at larger scattering angles ($>50^0$) can be seen.

Figure 8 shows diurnal variability of SSA retrieved at 440 nm for SZA greater than $50^0$, a limit which is a key quality control for Level 2 almucantars (Dubovik et al., 2000a; Holben et al., 2006). The results are presented for three different

combinations of SORD and inversion inputs listed in the figure captions. As is seen from the figure, the SSA retrievals by both algorithms under the same conditions (scalar RTC) exhibit noticeable SZA dependence. Switching to the vector mode in V3 retrieval code reduces the SZA dependence of SSA retrievals in both morning and afternoon. This can be explained by the higher accuracy of vector RTC in situations with substantial contribution of molecular scattering. In this particular Hamim case, the Rayleigh optical depth at 440 nm comprises $\sim 32\%$ of total optical depth which together with strong polarization of

molecular scattering makes accurate accounting for polarization important.

Fig. 9 shows Level 2 SSA retrievals by the V3 aerosol retrieval algorithm with RTC in vector mode for the three different types of input files described in the figure captions. Figure 8 shows a near constant offset in the values of retrieved SSA in the morning versus the afternoon which averages $\sim 0.014$ for 440 nm.  One of the possible explanations for this difference is that the dust in the morning and the afternoon was transported to Hamim from different regions. This assumption

is supported by the analysis of GSFC back trajectories for Hamim site (https://aeronet.gsfc.nasa.gov/cgi-bin/bamgomas_interactive), which shows different back trajectories in the afternoon versus morning.  Figure 9a shows that the contribution from the changes in NSR dominates that of BRDF for 440 nm SSA retrievals while according to Fig. 9b, the contributions from changes in NSR and BRDF at 675 nm are comparable. This is consistent with the results shown at Fig. 2, 4 and 6 which show that the changes in the temperature correction dominate 440 nm NSR while the 675 nm NSR are mostly

affected by the changes in solar flux and BRDF.   For SSA retrievals at 870 nm the contribution of the changes in BRDF is dominant, as shown at Fig. 10a, versus comparatively smaller effects for both solar flux and temperature correction.  The SSA retrievals at 1020 nm, shown at Fig. 10b, are affected primarily by the changes in temperature correction and BRDF, however acting significantly in opposite directions.



Figure 10 shows higher variability of SSA retrievals with SZA in the afternoon than in the morning for the inversions
employing both NSR and BRDF of V3. In particular, retrieved SSA values in the near - infrared wavelengths in the morning
are all close to 0.99 which is close to the theoretical limit for SSA of 1.0.  This can be explained by low dust absorption at
longer wavelengths (Dubovik, 2002) and constraints on the variability of retrieved parameters employed in the AERONET
inversion algorithm (Dubovik and King, 2000b). These constraints are based on a priori assumed ranges of variability of
aerosol parameters and impose limits on the values of retrieved parameters. Theoretically, the highest retrieved SSA value is
limited by 1, but because SSA is not retrieved directly (Dubovik and King, 2000b), the limits on SSA retrievals are imposed
through the constraints on the imaginary part of the refractive index (IRI). For low absorbing aerosols the actual IRI values are
very small and therefore even a small bias in simulated NSR can cause the retrieval values hit the assumed limit.  In the
morning, the IRI retrievals assume the AERONET constrained lower limit value of 0.0005 in which case the corresponding
SSA retrievals are concentrated around 0.99. In the afternoon, the lower IRI limit is not reached due to the higher aerosol
absorption.

### 3.4 Comparison of aerosol parameters retrieved by V2 and V3 retrieval algorithms.

This comparison of operational almucantar retrievals of V2 versus V3 is presented for four selected AERONET sites
(Mezaira, GSFC, Mongu, and Kanpur) and the results are summarized in the tables 1 through 17. For SSA and the real part of
refractive index the analysis is done for three bins of 440 nm AOD which allows for extending the analysis of the inversion
comparisons to different levels of sensitivity relative to the combined effects of measurement noise and modeling biases. For
the size distribution parameters two AOD bins (smaller and greater than 0.2 at 440 nm) are used due to the significantly higher
stability and accuracy of the aerosol size distribution retrievals (Dubovik, 2000a, 2002). The tables display statistics (mean
value and standard deviation) of the difference between aerosol parameters retrieved by the V2 and V3 retrieval algorithms.
For Mezaira, the SSA 440 nm retrievals by V2 and V3 are not biased with respect to each other. However, SSA retrieved
by V3 are systematically higher than those of V2 at longer wavelengths with the mean difference at 675 nm being the largest
due to cumulative effects of both solar flux and BRDF changes. Also, the mean differences are smaller for bins corresponding
to larger AOD. The mean difference of the retrievals of the real part of refractive index shows similar patterns at 440 and 675
nm but further increases reaching a maximum at 1020 nm. The agreement in size distribution parameters for both modes is
reasonable bearing in mind the mean climatological values for VMR and STD (in parentheses) for Mezaira site which are 0.15
μm (0.5) for fine and 2.3 μm (0.62) for coarse modes respectively. This translates into an average relative differences of only
~0.1% for VMRC and ~1.5% for STDC for AOD smaller than 0.2 and into ~0.8% and ~0.5% for greater AOD. The
corresponding numbers for the fine mode are also small: 5.4% and 0.5% for smaller AOD bin and 4.9% and 0.05% for AOD
larger than 0.2.
For GSFC, Mongu and Kanpur sites the mean differences between V2 and V3 in SSA retrievals at the two longer
wavelengths are much smaller than those for Mezaira due to smaller BRDF differences. However, the effect of the solar flux
on SSA 675 nm retrievals is much weaker for GSFC than that for Mongu and Kanpur, which explained by the difference in



the magnitude of aerosol absorption. To gain better understanding of the influence of the aerosol absorption on the uncertainty in retrieved SSA the formula (10) of Dubovik, et al. (2000a) can be used. Below this formula is rewritten to make dependence on SSA explicit:

$$\Delta\omega_0 \approx \omega_0 \left( \frac{\frac{\Delta\tau_{sca}}{\omega_0} - \Delta\tau_{ext}}{\tau_{ext}} \right), \tag{1}$$

where $\omega_0$ is SSA, $\tau_{ext}$ and $\tau_{sca}$ are extinction and scattering optical depth respectively, $\Delta\tau_{sca}$ and $\Delta\tau_{ext}$ are their uncertainties. Both uncertainties are always present: $\Delta\tau_{ext}$ is due to the accuracy of the AOD measurements and $\Delta\tau_{sca}$ may result from the modeling bias. For example, the changes in the magnitude of sky radiances due to the adjustment in solar flux value will be interpreted by inversion algorithm as a change in scattering optical depth. Eq. (1) shows that the contribution of the $\Delta\tau_{sca}$ to the SSA uncertainty depends on aerosol absorption: the smaller the aerosol absorption (larger $\omega_0$) the smaller the $\Delta\tau_{sca}$ contribution and vice versa. Figure 11 shows the results of simulation of the SSA uncertainty using Eq. (1) for the different levels of aerosol absorption (SSA of 0.99, 0.96, and 0.86) and gives a qualitative picture of the dependence of SSA uncertainty on aerosol absorption.

The parameters of size distribution retrieved by V2 and V3 are in good agreement for all three sites, which can be seen from the comparison V3-V2 mean differences to the following mean climatological values of VMR and STD (in parentheses) from all almucantar retrievals at each site: 0.166 μm (0.456) and 2.921 μm (0.672) for GSFC fine and coarse modes respectively, 0.146 μm (0.420) and 3.432 μm (0.655) for Mongu, 0.172 μm (0.486) and 2.85 μm (0.626) for Kanpur. The comparison gives the following average relative differences for VMR and STD (in parentheses) for the smaller AOD bin: 0.9% (4%) and 0.9% (1.9%) for GSFC fine and coarse modes respectively, 5.4% (4.2%) and 3.7% (0.8%) for Mongu, 0.4% (4%) and 3.3% (0.65%) for Kanpur. For the AOD greater than 0.2 the corresponding differences are the following: 3.7% (0.2%) and 2.1% (1.3%) for GSFC, 0.9% (0.5%) and 3.8% (1.7%) for Mongu, 0.9% (1.5%) and 4.7% (2.6%) for Kanpur.

The mean difference in retrieved real part of the refractive index is almost spectrally constant for Mongu site and decreases with increasing of wavelength for GSFC and Kanpur. However, in all the three cases, the standard deviations of the retrievals difference are significantly smaller than the ranges of operational almucantar retrieval variability for cases when AOD (440)>0.4 in both V2 and V3.

## 4. Hybrid scan: concept and retrieval scan.

As discussed in section 2, the reduction in the range of scattering angles of measured sky radiances affects the accuracy of SSA retrievals making them less stable in the presence of random noise, instrumental offsets, and modeling biases. Before V3, the two scan geometries utilized by CIMEL instruments were almucantar (ALM) and solar principal plane (PP) (Holben et al., 1998). The ALM scan is performed at a fixed view zenith angle equal to the SZA with varying azimuth angle ranging from $\pm 3.5^0$ to $\pm 180^0$, including sweeps in both directions from the sun position. The symmetry between these two ALM sides is used for the quality control of sky radiances measurements to screen out cloud cover and very inhomogeneous aerosol cases



(thick plumes). The maximum acceptable difference between the corresponding equal-angle almucantar scan measurement
pairs is 20% (Holben et al., 1998), measurements that exceed this threshold are eliminated. The range of scattering angles for
ALM geometry is the function of SZA and it is twice the value of the SZA thus making retrievals at small SZA less reliable
and/or biased (see Fig 7a). Level 2 criteria for SSA retrievals require SZA to be larger or equal to $50^0$ thus making utilizing
ALM retrievals for satellite products validation problematic at many latitudes and seasons (especially summer) due to the

relative closeness of satellite overpass time to local noon. In the PP scan geometry, the measurements are taken in the solar
principle plane (sun azimuth position) with the view zenith angle changing and the range of scattering angles equal to the sum
of SZA and maximum view zenith angle which is set to $75^0$ corresponding to $15^0$ elevation angle. Thus the PP scan geometry
has an advantage over that of ALM for small SZA observations due to the larger range of scattering angles measured. However,
due to the lack of the PP scan symmetry the quality assurance, primarily cloud screening, of measured sky radiances becomes

a problem. For new Cimel Model-T instruments this principal plane scan is turned off in favor of the newly design hybrid
(HYB) scan. Note that only the newer Model-T instruments can perform the HYB scan due to the requirement of greater
instrument controller processing capacity to compute the complete set of angles and perform this scan.

The HYB scan has the same range of scattering angles as the PP scan and yet also possesses the ALM scan symmetry
as depicted in Fig. 12. The HYB scan begins from the Sun and then proceeds in steps in such a way that the scattering angle

monotonically increases at each step which requires simultaneous adjustments of both view and azimuth angles. After reaching
the elevation angle of 15° the scan further proceeds at fixed view angle by varying azimuth angle similar to ALM scan except
than the view angle is not equal to SZA. The combination of two scans in two different planes allows maximizing the range
of scattering angles, similar as in the PP geometry. The scattering angle range of the HYB scan is equal to 75° plus the SZA.
Therefore, for a HYB scan made at 25° SZA the scattering angle range of measurements is 100°, which is the same scattering

angle range that the ALM scan is capable of at 50° SZA. Table 13 shows the detailed geometry of HYB scan for three SZA
values: 30°, 60° and 75°.

**4.1 HYB scan extension of SSA retrievals to smaller SZA.**

Due to the increased range of scattering angles for the HYB geometry, the corresponding SSA retrievals are expected

to exhibit less SZA variability than those of ALM and to a much lower SZA. To test this assumption, the dependences of SSA
retrieved from HYB scans on SZA were generated for three AERONET sites by aggregating retrievals in five SZA bins (each
bin is $10^0$ wide centered at 30°, 40°, 50°, 60° and 70° SZA) and calculating the mean value and the standard deviation for each
bin. Figure 13 shows the dependencies of 440 nm SSA retrievals on SZA for Mezaira, Mongu Inn, and Kanpur sites. Figure
13a shows the mean SSA values for SZA 50° and larger are nearly constant but decrease by ~0.007 for smaller SZA. The

magnitude of the decrease, however, is much smaller than that observed in the Hamim case study from almucantar retrievals
where SSA drops by ~ 0.045 between 50° and 30° (see Fig 7a). Figure13c shows a similar SZA dependence for Kanpur with
mean SSA variability of ~ 0.003 for SZA larger than $40^0$ and a decrease by ~ 0.015 between the 30° and 40° angular bins. As
was discussed in analysis of the Hamim almucantar inversions, the reason for the decrease of retrieved SSA with decreasing



SZA can be attributed to the bias in the modeling of light scattering by non-spherical dust aerosols when scattering angle range
is limited, which therefore can be expected to influence SSA retrievals at dust dominated locations and smaller SZA. In this
regard, both Mezaira and Kanpur sites are affected by dust aerosols: Mezaira is a desert site dominated by coarse mode dust
and Kanpur has a significant seasonal dust presence. The aerosol loading at Mongu Inn, from another hand, is dominated by
spherical fine mode aerosols from seasonal biomass burning and therefore no SZA dependent bias in SSA retrievals is
expected. This is illustrated by Fig. 13b which shows that the SZA dependence of SSA for Mongu Inn does not exhibit any
significant trend.

      The dependencies of SSA retrievals on SZA for HYB scans for longer wavelength are shown at Fig. 14 through 16. The
general feature of the results for Mezaira and Mongu Inn is the absence of any clear SZA trend for SSA retrievals. For Kanpur
the SZA dependence at 440 nm (Fig. 13c) is gradually reversed as wavelength increases reaching the maximum $30^0$ to $40^0$
SZA difference of ~ 0.018 at 1020 nm. The stability of dust SSA retrievals at Mezaira at longer wavelength can be explained
by the strong BRDF effect in V3 for barren desert sites which counterbalances the effect of modelling bias. Additional analysis
revealed that the SSA trend reversal for Kanpur can be explained by changing in relative contributions of fine and coarse
aerosol component to the total AOD with SZA bins. For example, at 1020 nm, $30^0$ SZA bin is dominated by coarse mode AOD
which is ~ 0.3 while fine mode AOD is only ~ 0.085. At the same time, at $40^0$ SZA bin the coarse and fine mode contributions
are comparable: ~ 0.18 (coarse) and ~ 0.16 (fine) resulting in SSA at $30^0$ being larger than that at $40^0$ (e. g. Eck et al., 2010).

### 4.2 Hybrid versus almucantar scan - comparison of SSA retrievals.

      Typically, HYB and ALM scans are taken ~ 10-12 minutes apart for the SZA range of $50^0$ to $75^0$, therefore the
corresponding retrieval results are expected to be in relatively good agreement. Fig. 17 and 18 show the statistics of the
difference in SSA retrieved from HYB and ALM scans as a function of SZA. The statistics were generated using the data from
all AERONET sites for which the HYB inversions were available. In general, the agreement is good for all wavelengths with
the mean values of the differences below 0.005. At the same time the variability is increasing with increasing wavelength due
to predominant contribution of fine mode aerosols to the generated statistics, and therefore much smaller AOD at the longest
wavelengths which results in less sensitivity to aerosol absorption. Also there is a trend of decreasing of the difference with
increase of SZA due to increasing similarity of both scan geometries which is very clearly visible at 440 nm. At longer
wavelengths SZA dependencies exhibit maximum at ~ $55^0$ due to sharp decrease of the mean difference at the smallest SZA.

### 5     Uncertainty estimates of the retrieved aerosol parameters: U27

      The first assessment of the accuracy of aerosol parameters retrieved by AERONET sky scan inversions was presented
in (Dubovik, 2000a). This analysis was performed by sensitivity studies for a set of preselected aerosol models assuming given
input uncertainties due to random errors, instrumental offsets, and known uncertainties in atmospheric radiation modelling.
The main results are presented in Table 4 of that paper, summarizing the dependence of the accuracy on aerosol type for a
limited number of AOD values and for a single fixed value of solar zenith angle (60 °). Applicability of the results to the 50-



70° range of solar zenith angles is therefore assumed. The AOD dependence was analyzed for two AOD values for water-soluble aerosols (smaller and greater than 0.2 at 440 nm) and for one AOD range (greater than 0.5 at 440 nm) for biomass burning and dust aerosol types. The accuracy of wavelength dependent aerosol parameters (real and imaginary parts of refractive index and single scattering albedo) was only analyzed for 440 nm. Based on these results the following thresholds for level 2 aerosol retrieval quality control for the refractive indices are used: AOD (440) is greater than 0.5 and SZA is greater than 50° (Dubovik, 2000a; Holben et al., 2006). These thresholds were selected in order to insure that for Level 2 data the uncertainty of retrieved SSA was 0.03 or less. For the size distribution retrievals there was no lower limit threshold in 440 nm AOD to meet quality control for Level 2 data.

In V3 the analysis of the accuracy of retrieved aerosol parameters is extended to all the wavelengths participating in inversion (440, 675, 870 and 1020 nm) and to a wider range of values of AOD and SZA. The approach is somewhat similar to that of Dubovik (2000a) in that the variability in retrieved aerosol parameters is generated by perturbing observations in inversion inputs as in typical sensitivity studies. However instead of generating synthetic measurements for the prescribed sets of aerosol models and then perturbing them, the real time Sun photometer observations are perturbed by the assumed values of instrumental and auxiliary input biases. This allows avoiding some rather complex modeling such as considering aerosols of different types or mixtures thereof over different types of surfaces which would require a large set of different combinations to analyze. Also, a very wide range of AODs and SZAs are covered since the entire multi-decade data base of AERONET almucantar observations were analyzed in this V3 assessment of uncertainty.

## 5.1 Approach for individual inversions.

The assumption made in this analysis is that the perturbations in measurements and auxiliary inputs are due to uncertainties (biases, systematic errors) in AOD, radiometric calibration of sky radiances, solar spectral irradiance, and surface reflectance. Sun photometer pointing bias is not included in the list of uncertainties because this error is quite small (~0.1 °) and furthermore averaging the left and right parts of almucantar scans before use as input to the retrieval code reduces this uncertainty significantly. Each bias assumes three values: positive value, negative value, and zero (non-biased), and the values and the signs of biases of the same type are assumed spectrally independent. This means, for example, that AODs at different wavelengths are biased in the same direction with the same assumed bias value. The same is true for sky radiances, solar irradiance spectrum, and surface reflectance. The radiometric calibration and solar spectrum irradiance uncertainties are combined in one bias because both of them affect the magnitude of sky radiances. As a result, there are three distinct biases each assuming three values which makes 27 distinct combinations in perturbing measurements in inversion inputs. The following values for one sigma biases of the input data are assumed. AOD: $\pm 0.01/m$ (Eck et al., 1999b), where $m$ is optical air mass, radiometric calibration bias: $\pm 3\%$, solar irradiance spectrum: $\pm 2\%$., and surface reflectance: $\pm 0.05$. In Ross -Li BRDF model, only the first of the three BRDF parameters is perturbed. This is equivalent to an assumption that the angular shape of BRDF is unchanged and only the surface albedo is biased. For each set of Sun photometer observations an array of 27 inputs



corresponding to 27 combinations of biases are produced. All of these inputs are then separately inverted and the following statistics of the inversion results is generated: average value, standard deviation, minimal and maximal values. These statistics describe the variability in retrieved aerosol parameters due to the assumed uncertainties in inversion inputs for each individual inversion. From these statistics only the standard deviation (which we label as U27) is used as a proxy for the estimated

uncertainty.

## 5.2  Look Up Table (LUT) approach.

Because operationally running 27 inversions instead of one substantially increases the computational time, a LUT approach was designed to speed up the determination of U27 for all new observations when they are reprocessed to Level 2

with final calibrations applied. A climatological LUT is generated from the entire AERONET almucantar and hybrid scan database by binning U27s in AOD (440 nm), Angstrom Exponent (AE, 440- 870nm), and SSA (440, 675, 870, 1020 nm). A total of ~1 million sky scans that passed Level 2 quality controls were inverted 27 times in order to produce the U27 values for all parameters. Using this LUT approach, U27 for each individual new Level 2 retrieval can be obtain by interpolation using the corresponding measured and inverted combination of AOD, AE, and SSA. The analysis of U27 for dust aerosol

individual retrievals of SSA at longer wavelengths revealed physically nonrealistic small values of U27 related to IRI hitting its lower limit (boundary) constraint of the inversion as was discussed in analysis of the results presented on Fig. 10. Therefore, a second LUT was generated from the subset of U27s for only individual inversions when no boundary hits were detected for any of the retrieved parameters. The second LUT is used for obtaining the unbiased estimates of uncertainty in situations when the effect of boundary hitting by retrievals on U27 is substantial (typically for SSA of dust aerosol, which is already very

weakly absorbing).

The following binning intervals were used in generating the LUT.  AOD: less than 0.05, 0.05-0.1, 0.1-0.2, 0.2-3.4 (in 0.2 increment bins). AE (440-870 nm): less than 0.3, 0.3-2.4 (in 0.3 increment bins). SSA: 440, 675 nm: less than 0.7, 0.7-0.8, 0.8-0.85, 0.85-0.9, 0.9-0.95, 0.95-0.975, and 0.975-1.0. SSA: 870-1020 nm: two additional bins: less than 0.6, and 0.6-0.7. The U27 are provided for the following parameters: VMR of size distribution for both modes, width of size distribution for

both modes, IRI at four standard wavelengths, SSA at four standard wavelengths, and asymmetry parameter at four standard wavelengths. The uncertainty estimates for the real part of refractive index (RRI) are not provided for the reason that will be discussed later in section 5.4.2.

All parameters use AOD and AE in the LUT; however, some absorption dependent parameters also utilize the binning of SSA, which produces two types of LUTs. The first LUT is two dimensional with binning in AOD and AE. This LUT is

used to generate U27 for VMR, width of PSD, and asymmetry parameter. The second LUT is binned by AOD, AE, and SSA and used to obtain U27 for IRI, and SSA in order to account for the effect of different absorption levels by aerosols with overlapping AE ranges, e. g. pollution and smoke aerosols (both of which exhibit a wide range of absorption (Dubovik et al, 2002; Giles et al, 2012)). The two dimensional (2D) LUT can be viewed as a rectangular grid, each node of which corresponds





to an AOD, AE pair. Each inversion has its own AOD and AE, and thus represents a point in the AOD, AE 2D space. If a point
is inside of that grid, four surrounding nodes are used for 2D interpolation. Similarly, the three dimensional (3D) LUT can be
viewed as a parallelepiped lattice, each node of which corresponds to an AOD, AE, SSA triad. Each inversion has its own
AOD, AE and SSA, and thus represents a point in the AOD, AE, SSA 3D space. If a point is inside of the lattice, we use the
eight surrounding nodes for 3D interpolation. If one or more points that are required for interpolation are missing, the values
of the closest nodal points are used. The 2D interpolation is done on the original LUT (no restriction on boundary hits of the a
priori constraints). The 3D interpolation uses the no hits LUT (no constraint limits reached for any retrieved parameters) only
for the following combination of AE, SSA bins: AE– less than 0.3 and SSA–0.975-1.0, conditions for which the most of the
boundary hits of the IRI parameter occurs.

Figure 19 shows comparison of U27 estimated for SSA retrievals by the two methods for Mezaira for SSA at 440 nm
and 675nm. Figure 19a shows good agreement between them at 440 nm although the dependence of LUT interpolated U27
on AOD exhibits less scattering than that for individual inversions which can be explained by the additional LUT averaging.
Figure 19b at 675 nm demonstrates the effect of the boundary hits by IRI retrievals on U27 obtained for individual inversions
in which case the variability among the 27 individual retrievals is reduced due to the lower limit constraint for IRI resulting in
an upper limit for SSA. Because of this artificial variability reduction, the corresponding standard deviations are
underestimated in some of the individual retrievals. It is also seen from the Fig. 19b that combining the original LUT with the
one with no hits for the lowest AE and the highest SSA bins corrects these unphysically small U27 values especially for low
levels of AOD.

### 5.3 Quality control.

For V3 retrieval parameters, we define the Level 2 criteria for estimated retrieval uncertainties in a manner very
similar to that of Dubovik et al. (2000a) and Holben et al. (2006), thus limiting the quality assured U27 retrievals to those
within AOD/SZA boundaries close to or equal to those defined in the previous V2. The following Level 2 criteria are used and
U27 estimates of uncertainty are provided for these parameters:

- SSA, IRI: for ALM 440 nm AOD larger than 0.4 and SZA larger than 50°. For HYB 440 nm AOD larger than 0.4
and SZA larger than 25°. This insures that for both ALM and HYB scans the minimum angle of scattering angle range
is 100° for measured directional sky radiances.
- Volume median radius, width of particle size distribution, asymmetry parameter: For ALM 440 nm AOD larger than
0.02 and solar zenith angle (SZA) larger than 50°. For HYB 440 nm AOD larger than 0.02 and SZA larger than 25°.

### 5.4 Analysis.

The approach utilizing LUT interpolation was investigated by generating U27 for aerosol retrievals at four selected
AERONET sites: GSFC, Mezaira, Kanpur, and Mongu for which the results are presented below.





### 5.4.1    SSA

Figure 20 shows the uncertainties estimates for SSA retrieved at the GSFC site. As is seen, at all the wavelengths the U27 exhibits power like functional dependencies on AOD with smaller uncertainties corresponding to the larger optical depth

and vice versa. The scattering of the data increased with increasing wavelength which can be explained by the decrease in AODs at longer wavelengths due to the large AE, 1.61 yearly climatological value. In addition, the data displayed in Fig. 20 are comprised of retrievals for the range of SZA greater than $50^0$ (upper limit $\sim77^0$) which can contribute to additional U27 scattering due to SZA variability within narrow AOD bins. The solid blue lines on Fig. 20 indicate the upper boundary of Level 2 SSA retrieval uncertainty established in Dubovik et al. (2000a) for AOD (440 nm) greater than 0.2 for water-soluble

aerosol type (weakly absorbing) which dominates GSFC aerosol loading especially at the highest AOD levels. The close inspection of Fig. 20a shows that the crossing point between the blue line (Dubovik et al. (2000a) estimate) and U27 AOD dependence is close to AOD (440) of 0.3 thus increasing the AOD by 0.1 in V3 for the same estimate of SSA uncertainty (0.03) that was assumed in V2 (for water soluble aerosol). Similar analysis of Fig. 20b though Fig. 20d show that the AOD (440 nm) corresponding to the uncertainty of 0.03 increases with wavelength: $\sim$ 0.4, 0.5, 0.6 for 675, 870, and 1020 nm

respectively.

Figure 21 shows the uncertainty estimates for SSA retrieved at the Mezaira site. The uncertainties exhibit some minor scattering at 440 nm and much larger variability at longer wavelengths. The main reason for the larger scattering of U27 values at longer wavelengths is the close proximity of the original IRI retrievals to the lower limit set by the inversion code even after the direct boundary hits were filtered out, resulting in small uncertainty values near the limit. In this case the different degree

of proximity to the lower limit results in different degree of reduction of U27 thus producing the resulting distribution of U27 values for each narrow AOD bin. The closeness of the IRI retrievals to the inversion limit at longer wavelength in V3 is partially due to the BRDF input in V3 as discussed previously.  The U27 exhibits smaller scattering for AOD larger than $\sim$0.9 which can be explained by the higher sensitivity to aerosol absorption in this AOD range as well as by the smaller number of the occurrences of high AOD events. Additionally, as AOD increases the aerosol signal increases relative to the fixed

uncertainties of data and auxiliary inputs. Figure 21a shows the 0.03 uncertainty level corresponds to the 440 nm AOD value of $\sim$0.45 which is close to 0.5 value by Dubovik et al. (2002) which gave the same AOD value for both dust and biomass burning aerosols. At longer wavelengths, the intersection point of the 0.03 line with U27 AOD dependence is harder to define due to fair amount of scattering, however, intersection with upper boundary of U27 AOD dependence is in vicinity of 0.45 as well. However, it may be somewhat less than AOD (440) =0.45 due to the fact that absorption by dust is very weak at the

longer wavelengths therefore resulting in greater sensitivity to absorption while it is much stronger at 440 nm where absorption occurs from the iron oxides in dust.

The dependence of estimated retrieval uncertainties on AOD can be made more explicit by averaging the scattering of individual data points over narrow AOD bins. Following this reasoning, the U27 SSA data for each of the four representative AERONET sites were binned in AOD with the number of bins selected in such a way as to cover the widest AOD range





possible and at the same time keep the number of data points in each bin high enough for representative statistics. Then the average values for each AOD bin were calculated and the final U27 dependencies were obtained by interpolating the binned data to a regularly spaced AOD grid. Fig. 22a through Fig. 22d show the resulting AOD dependencies of SSA uncertainties for each of four selected AERONET sites. The corresponding numerical data are summarized in Tables 14 through 17 for AOD (440 nm) less than 0.7. Figure 22 shows that as expected AOD averaging resulted in a rather smooth AOD dependencies

of U27. The graphs of Fig. 22 provide a clear visualization of the spectral dependence of SSA uncertainties by considering intersections of corresponding curves with the 0.03 uncertainty level. For fine mode aerosols dominating both the GSFC and Mongu sites, the values of AOD corresponding to 0.03 SSA uncertainty are increasing with the wavelength as can be seen from Fig. 22a and Fig. 22c. In the GSFC case, for example, the 0.03 accuracy for 1020 nm SSA is reached at AOD of ~0.6 as compared to AOD of ~0.3 for 440 nm SSA. For coarse mode aerosols such as mineral dust (Mezaira site), the spectral

variability of SSA uncertainty is much weaker as is seen from the graphs of Fig. 22b. The decrease in U27 of ~0.0085 for longer wavelengths relative to 440 nm can be explained in part by the underestimation of uncertainty due to the proximity of IRI to the inversion limit as was previously discussed. In addition, the lower SSA uncertainties can be related to the lower dust absorption in this spectral range as was discussed in relation to Eq.1. The aerosol loading over Kanpur is representative of the mixture of fine mode pollution and coarse mode dust with dust aerosols dominating during April through June. The effect of

different components of the mixture on AOD dependence of U27 is clearly seen in Fig.22d. For example, for the AOD larger than ~ 0.8 the contribution of fine mode aerosols is dominant resulting in clear separation of U27 values at different wavelengths. For the range of smaller AOD the dust contribution is stronger as can be concluded from the much weaker spectral dependence of estimated SSA uncertainties.

**5.4.2    Refractive index**

The behavior of the U27 uncertainty estimates for the IRI are very similar to that of SSA. Fig. 23 shows examples of uncertainties for 440 nm IRI estimated at GSFC and those of 1020 nm IRI estimated at Mezaira. There is a clear similarity between AOD dependences of IRI uncertainties of Fig. 23 and AOD dependencies of uncertainties of the corresponding SSA, see Fig. 20a and 21d. The difference in the average magnitude of U27 estimates of GSFC and Mezaira is due to the very low

dust absorption at 1020 nm as well as the effect of the lower boundary retrieval constraint for IRI.

Fig. 24 shows the uncertainties (by the U27 methodology) of the RRI at 440 nm estimated for the GSFC, Mongu, and Mezaira sites. The values of these uncertainties at the GSFC and Mongu sites are below 0.02 for AOD greater than 0.4 and further decrease with AOD reaching minimum value of ~ 0.012 while the U27 estimated at Mezaira level out at ~ 0.021 for AOD greater than ~ 0.30. The average U27 were calculated for the range of AOD larger than 0.5 and compared to the values

reported in (Dubovik et al, 2000a) for different aerosol types: 0.02 (Dubovik: 0.04) for dust at Mezaira, 0.015 (Dubovik: 0.04) for biomass burning at Mongu, and 0.0175 (Dubovik: 0.025) for water soluble at GSFC. The difference in U27 estimates and uncertainties of (Dubovik et al., 2000a) can be explained by the different way of accounting for instrument pointing bias. As




was discussed before, V3 uncertainty estimating does not account for pointing because of its much reduced effect due to the averaging of left and right parts of the ALM or HYB scans. On the other hand, a pointing bias of $0.5°$ was assumed in the sensitivity analysis of Dubovik et al. (2000a). This assumed value of the pointing bias is in excess of a more realistic value of $0.1^0$ which was estimated by comparing the two sides of ALM scans. Thus additional unrealistically large source of uncertainty was introduced in Dubovik et al. (2000a) with potentially large effects on the RRI retrievals. Further analysis showed that U27 estimates are substantially lower than observed variability (standard deviations) of RRI retrievals: ~ 0.05 at Mezaira, ~ 0.06 at Mongu, and ~ 0.049 at GSFC. In contrast, the U27 estimated for SSA and observed variability of SSA retrievals are more consistent. Additionally, the upper and lower limit retrieval constraints of RRI (1.33 and 1.60, respectively) were reached at times, suggesting some potential instability in the retrievals of RRI. The percentage values of RRI retrievals reaching 1.33 and 1.6 (in parentheses) were estimated for all three sites for 440 nm AOD less than 0.2 and greater than 0.4. The estimates for smaller AOD bin show that RRI retrievals hit 1.6 boundary more often than that of 1.33: 1.6% (13.7%) for Mezaira, 3.1% (11.1%) for GSFC, and 2.2% (18.5%) for Mongu. The corresponding estimates for larger AOD demonstrate significant reduction in the number of boundary hits, especially that of 1.6: 0.6% (6.8%) for Mezaira, 2.5% (0.9%) for GSFC, and 0.4% (4.3%) for Mongu.

The apparent underestimation of RRI uncertainties by U27 can be related to additional factors other than uncertainties in inversion inputs which can influence retrieval variability. Figure 25a shows the dependence of original retrievals of RRI at 440 nm on the volume concentration of the fine mode (CVF) derived from the retrieved particle size distribution for the GSFC site data (nearly all fine mode dominated). For this plot the data were sampled into five AOD bins which resulted in separating the RRI dependence on CVF into five distinct branches with a rather strong correlation between RRI and CVF. Each branch is well approximated by linear dependence with regression equations and correlation coefficients displayed in the figure. As is seen the absolute values of the slopes of linear regressions are decreasing as AOD increases. Simultaneously, the range of the variability of CVF increases and that of RRI decreases as AOD increases. The strong variability of RRI retrievals for narrow AOD bins can be explained by the lack of sufficient measurement sensitivity to this parameter to provide stable retrievals under somewhat noisy measurement conditions at times. In particular, AOD measurements do not constrain RRI individually but in conjunction with CVF as it follows from results of Fig. 25a. As RRI is changing it affects extinction optical depth though the change in aerosol scattering. However, to keep modeled extinction optical depth consistent with the AOD measurements CVF is adjusted by the inversion code to compensate for the change caused by RRI. Thus decreases in RRI are compensated by increases in CVF and vice versa in such a way as to reproduce the total aerosol scattering. Therefore, accurate retrievals of SSA are possible in spite of the variability of individual RRI and CVF retrievals, since total aerosol scattering is accurately described by a combination of these two parameters. The effect of the correlated behavior of RRI and CVF retrievals can be seen in Fig.25 b where it manifests itself in the scattering of CVF retrievals for the narrow AOD bins. Similar analysis was done for Mongu site which is also dominated by fine mode aerosol. As in GSFC case the RRI vs CVF anti- correlation was observed which allows extending the above conclusion to all fine mode dominated aerosols.





In the case of dust aerosols, AOD measurements do not provide sufficient constraints for RRI retrievals due to the very small sensitivity of AOD to variability in RRI for these coarse mode size particles. This is a consequence of a well-known fact that the extinction efficiency factor of large particles tends to a constant limit as particle size increases (e. g. Bohren and Huffman, 1998). In both cases, fine and coarse aerosols, constraints provided by the measurements of sky radiance intensities are not sufficient to provide stable RRI retrievals under conditions of typical measurement noise. The addition of polarization measurements can increase the sensitivity to RRI as demonstrated in Fedarenka et al. (2016). The U27 for individual inversions are estimated under the specific realization of measurement noise which equally affects the original retrieval as well as retrievals obtained by inverting perturbed input files. Thus, the U27 estimates are not sensitive to the variability of retrievals caused by slightly noisy measurement conditions and will underestimate the retrieval uncertainty for RRI due to lack of sufficient measurement constraints. For this reason, the U27 estimates for the RRI are not reported in the AERONET database.

### 5.4.3    Size distribution

Figures 26a and 26b show estimated uncertainties (U27) for volume median radius for fine mode (VMRF) and standard deviation (width) of the size distribution for the fine mode (STDF) plotted as a function of AOD for the GSFC site. Uncertainties for both parameters are decreasing as AOD increases reaching ~ 0.004 and ~ 0.009 for VMRF and STDF respectively for high AOD (>0.4 at 440 nm). Comparison of the U27 values from all L2 retrievals averaged over the entire AOD range to the variability of retrievals (in parentheses) shows that the U27 values are smaller than those of the standard deviation of retrieval variability: 0.008 (0.03) for VMRF and 0.02 (0.06) for STDF.  This difference can be explained by the actual variability of VMRF and STDF due to the different levels of humidification growth, cloud processing and/or particle aging (Eck et al., 2012). Averaging over an AOD range greater than 0.4 results in decreasing the U27 values but does not produce significant changes in the standard deviation of retrievals: 0.0043 (0.032) for VMRF and 0.009 (0.056) for STDF. The VMRF and STDF uncertainties (U27) estimated at the Mongu site are shown in Fig.27a and Fig.27b respectively and exhibit similar behavior with AOD as the uncertainties estimated at the GSFC site. Similarly, the U27 values averaged over the entire AOD range are significantly smaller than the standard deviations of retrievals variability: 0.0058 (0.0207) for VMRF and 0.0138 (0.0616) for STDF. Averaging over AOD greater than 0.4 reduces both U27 and standard deviation values: 0.0039 (0.0118) for VMRF and 0.0094 (0.0416) for STDF. Larger values of standard deviation at larger AOD can be explained by natural variability in aerosol size. For the Mongu site the average value of Angstrom exponent (440 – 870 nm) is ~ 1.67 over the entire AOD range with standard deviation of ~ 0.24. We conclude that variability in aerosol size can contribute to the observed variability of aerosol retrievals (Eck et al. 2001, 2013).

Uncertainties (U27) in the size distribution parameters for the Mezaira site are presented in Fig. 28 for volume median radius and size distribution standard deviation for coarse mode (VMRC and STDC) since this is a desert dust dominated location. The VMRC dependence on AOD follows the pattern similar to that of fine mode aerosols with U27 values averaged over entire AOD range are significantly smaller than standard deviations of L2 retrieval variability: 0.073 (0.356) and 0.068



(0.356) for averaging over AOD larger than 0.4. The ratio of VMRC U27 to the standard deviation of VMRC retrievals
averaged over 440 nm AOD greater than 0.4 is ~ 3.2%. A significant amount of scattering can be seen from Fig.28b for STDC
uncertainties for AOD smaller than 0.5. This can be attributed to the presence of significant fraction of smaller particles for
AOD smaller than 0.5 for which the coarse mode is not well defined. This also partly explains the observed larger variability
in retrieved size distribution parameters at low AOD.

For the Kanpur site, the U27 estimates of uncertainty for size distribution parameters are presented for both fine and
coarse modes in Fig. 29 and Fig. 30 respectively. As is shown in Fig.29 the U27 exhibits two populations, with one displaying
a strong increase in U27 values between AOD values of 0.5 and 1.0. This increase can be explained by the presence of dust
aerosols with strong coarse mode for which therefore the fine mode AOD is relatively low thereby providing low information
content for the fine mode in the retrieval which results in greater sensitivity to the uncertainties in the inversion inputs.
Similarly, the U27 increase for coarse mode radius shown in Fig.30 for AOD larger than one is related to the domination of
fine mode aerosol with relatively low coarse mode AOD and therefore weak signal in the measurements attributed to the coarse
mode. The U27 values averaged over entire AOD range and for AOD larger than 0.4 are very close to each other and have the
following values for the average of the entire AOD range, along with standard deviations of L2 retrieval variability: 0.0063
(0.0465) for VMRF, 0.013 (0.066) for STDF, 0.0753 (0.405) for VMRC, and 0.0187 (0.0538) for STDC.

**6    Summary and conclusions.**

The changes and additions to the V3 aerosol retrieval algorithm over that of V2 are presented and discussed. They
include: a new polarized radiative transfer code SORD which replaced the scalar code of V2, detailed characterization of gas
absorption by adding  $NO_2$ and $H_2O$ to specify total gas absorption in the atmospheric column, specification of vertical profiles
of all the atmospheric species, new BRDF parameters for land sites adopted from the MODIS BRDF/Albedo product, new
version of the extraterrestrial solar flux spectrum, and new temperature correction procedures of both AOD and sky
measurements.  The hybrid inversion processing is introduced and the uncertainty product (U27) is provided for specific
inversion products.

The potential effect of each change in V3 on aerosol retrievals is analyzed.  It is shown that desert BRDF of V3 is less
anisotropic than that of V2, sometimes resulting in large differences in surface albedo between two versions. The effect is
important at longer wavelengths and especially at high SZA and can adversely affect the desert dust SSA retrievals at 675,
870, and 1020 nm channels. For non-desert sites the agreement in surface reflectance between V2 and V3 is reasonable. The
comparison between the extraterrestrial solar fluxes of V2 and V3 revealed an anomalously large relative difference of ~ 3%
at 675 nm which potentially affects the 675 nm SSA retrievals for all aerosol types. Also, it is demonstrated that new
temperature correction procedures can substantially affect normalized sky radiances for instruments using older models of
filters.

The combined effects of the above factors on retrievals of SSA were analyzed in a case study for August 25, 2004 at
Hamim which is desert site in United Arab Emirates. Analysis was done by making sequential changes in input files starting



from V2 inputs and observing corresponding changes in retrieved SSA values. It was shown that the 440 nm SSA retrievals were primarily affected by the new temperature correction procedure due to similarity of V3 BRDF and that of V2 and smaller

than 1% difference in extraterrestrial solar flux. The 675 nm SSA retrievals are influenced by the extraterrestrial solar flux and BRDF effects which were additive and the 870 nm SSA changes were almost solely due to the changes in BRDF with very low temperature sensitivity at this channel and extraterrestrial solar flux difference close to zero. At 1020 nm channel SSA retrievals are predominantly impacted by temperature correction and BRDF. These results, while derived from one case study only, could be used as a general guideline in analyzing SSA retrievals in some desert environments.

Operational almucantar retrievals of V2 versus V3 were compared for four AERONET sites: GSFC, Mezaira, Mongu, and Kanpur. The results are presented as statistics (mean value and standard deviation) of the difference between aerosol parameters retrieved by the V2 and V3 retrieval algorithms. For size distribution, mean differences are compared to climatological means for each selected site and percentage values of the ratios are calculated for VMR and STD for both fine and coarse modes. In general, the percentage values vary between ~0.1% and ~ 5.4% for VMR and ~0.5% and 4% for STD

which indicates a reasonable good agreement in size distribution retrievals. The SSA comparisons are affected by all those factors discussed earlier in relation to the case study. For example, for the desert dust site, Mezaira, the comparison shows that SSA retrieved by V3 at longer wavelengths are systematically higher than those of V2 with the mean difference at 675 nm being the largest due to cumulative effects of both solar flux and BRDF changes. For non-dust sites the main discrepancy is between 675 nm SSA retrievals due to extraterrestrial solar flux difference. The mean SSA differences at 675 nm, however,

are very different for low and high absorbing aerosols, for example, 0.0007 for GSFC versus 0.01 for Mongu. This is explained by the lower sensitivity to a bias in aerosol scattering optical depth by more weakly absorbing aerosols.

     A new sky radiance angular distribution measurements scan, called the hybrid (HYB), is introduced and discussed. The HYB combines scans in two different planes to maximize the range of measured scattering angles while also achieving scan symmetry which aids in data quality control, especially cloud screening. The HYB scan proceeds in steps in such a way that

the scattering angle monotonically increases at each step which requires simultaneous adjustments of both view and azimuth angles. This is in contrast to ALM and PP scans where just one of the angles is changed. It has been shown that due to the extended range of scattering angles at smaller SZAs, the HYB retrievals of SSA for dust aerosols exhibit smaller variability as a function of SZA than those of ALM. For example, the average decrease of 440 nm HYB SSA retrievals between $50^0$ and $30^0$ SZA is only ~ 0.007 whereas in the case study for Hamim the corresponding SSA decrease for the ALM scan is ~ 0.045. The

SZA variability of dust SSA retrievals manifests itself as a decrease in SSA retrievals with decreasing SZA which is explained by the SZA dependent bias in the model describing single scattering properties of non-spherical dust aerosols. The comparison of SSA retrievals from successive HYB and ALM scans in the $50^0$ to $75^0$ SZA range (range of scattering angle measurements >100 degrees) showed a good agreement with the differences below ~ 0.005.

     A new approach to estimate uncertainties in the retrieved aerosol parameters was developed. It uses variability in the

inversion retrievals of aerosol parameters generated by perturbing both input measurements and auxiliary input parameters as a proxy for retrieval uncertainty. The perturbations in measurements and auxiliary inputs are due to estimates of biases in



AOD, radiometric calibration uncertainty of sky radiances combined with uncertainties of solar spectral irradiance, and surface bidirectional reflectance. Each bias assumes three values: positive, negative, and zero (non-biased), and the values and the signs of biases of the same type are assumed spectrally independent which makes for 27 distinct combinations of perturbations

to the inversion inputs. For each set of Sun photometer observations 27 inputs corresponding to these 27 combinations of biases were produced and separate inversions performed for each. The following statistics of the 27 individual inversion results were generated: average, standard deviation, minimum and maximum values. These statistics describe the variability in retrieved aerosol parameters due to uncertainties in inversion inputs for each individual inversion. From these statistics the standard deviation (labelled as U27) is used as a proxy for estimated retrieval parameter uncertainty.

815        Due to substantially increased computational time in running 27 inversions instead of one, a LUT approach was designed to speed up the determination of U27 for all new observations. A climatological LUT was generated from the entire Level 2 AERONET almucantar and hybrid scan database by binning U27s in AOD (440 nm), Angstrom Exponent (AE, 440-870nm), and SSA (440, 675, 870, 1020 nm). Using this LUT approach, U27 values for each individual new Level 2 retrieval can be obtained by interpolation using the corresponding measured and inverted combinations of AOD, AE, and SSA. There

are two types of LUT. The first LUT is two dimensional with binning in AOD and AE. This LUT is used to generate U27 for VMR, width of PSD, and asymmetry parameter. The second LUT, three dimensional, is binned by AOD, AE, and SSA and used to obtain U27 for IRI, and SSA in order to account for the effect of different absorption levels by aerosols with overlapping AE ranges, e. g. pollution and smoke aerosols. To prevent underestimation of U27 due to retrieved aerosol parameters hitting a priori boundaries set up by the inversion code, the second 3D LUT was generated from subset of U27s for only those

individual inversions when no boundary hits were detected for any of the retrieved parameters. This LUT is used for obtaining the unbiased estimates of uncertainty in situations when the effect of boundary hitting by retrievals on U27 is substantial.

        The LUT approach was tested by generating U27 for aerosol retrievals at four selected AERONET sites: GSFC, Mezaira, Kanpur, and Mongu (representing differing aerosol types) and analyzing dependencies of U27 on 440 nm AOD. For SSA, it is found that U27 estimates are typically consistent with uncertainties reported in (Dubovik et al., 2000a). For example,

the 440 nm AOD values corresponding to the V2 upper boundary for 440 nm SSA uncertainty (0.03) are close to those of (Dubovik et al., 2000a (in parentheses): 0.3 (0.2) at GSFC and 0.45 (0.5) at Mezaira. For longer wavelengths, the 0.03 uncertainty is reached at higher 440 nm AODs for fine mode aerosols, due to significantly lower AOD at longer wavelengths, and at similar AOD for dust, due to weak wavelength dependence of AOD. The smoothed dependencies of SSA uncertainties on 440 nm AOD for selected AERONET sites were generated by averaging of U27 over narrow AOD bins. The resulted data

are tabulated for each site.

        The U27 estimated for RRI are smaller than those of Dubovik et al. (2000a) and significantly smaller than the observed variability of operational retrievals. The difference in U27 estimates and uncertainties of (Dubovik et al., 2000a) is explained by differences in estimating the magnitude of instrument pointing bias. The U27 estimates do not account for pointing because of analysis showing small pointing bias plus its much reduced effect due to the averaging of left and right

parts of the ALM or HYB scans, while an unrealistically large pointing bias of $0.5^0$ was assumed in the sensitivity analysis of





Dubovik et al. (2000a). It is shown that U27 underestimates RRI uncertainty due to insufficient measurement constraints, when polarization measurements are not available. The effect of variability in RRI on aerosol scattering for fine mode aerosols is partially counterbalanced by variability in volume concentration which is anti-correlated with RRI. This allows accurate reproduction of aerosol scattering and correspondingly accurate retrieval of SSA, while resulting in significant uncertainty in

RRI and fine volume concentration. For dust aerosols AOD is not sensitive to RRI due to asymptotic behavior of extinction and scattering by large aerosol particles. Thus, the U27 of RRI estimates are not sensitive to the variability of retrievals caused by noisy measurement conditions and will underestimate the retrieval uncertainty for RRI due to lack of sufficient measurement constraints. For this reason, the U27 estimates for the RRI are not reported in the AERONET database.

     The U27 estimates for size distribution (VMR and STD) averaged over the entire AOD range are compared to the

variability of operational retrievals. In general, the averaged U27 are smaller than standard deviations of retrievals variability for both VMR and STD. The difference can be explained by natural variability of aerosol size distribution at selected sites. The estimates of uncertainty (U27) in size distribution VMR and STD parameters are relatively small even for low AOD, therefore these parameters are designated as Level 2 quality for nearly all AERONET retrievals, when AOD at 440 nm>0.02.

Acknowledgements:

The AERONET project at NASA GSFC is supported by the Earth Observing System Project Science Office cal–val, Radiation Sciences Program at NASA headquarters, and various field campaigns. Resources supporting this work were provided by the NASA High-End Computing (HEC) Program through the NASA Center for Climate Simulation (NCCS) at Goddard Space Flight Center.

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





| AOD(440 nm) | SSA (440 nm) | SSA (675 nm) | SSA (870 nm) | SSA (1020 nm) | Number of retrievals |
|---|---|---|---|---|---|
| > 0.4 | 0.001 (0.008) | 0.020 (0.009) | 0.016 (0.009) | 0.012 (0.010) | 1473 |
| 0.4-0.6 | 0.002 (0.008) | 0.022 (0.008) | 0.017 (0.009) | 0.013 (0.011) | 1012 |
| > 0.6 | -0.001 (0.006) | 0.015 (0.007) | 0.012 (0.007) | 0.010 (0.007) | 461 |

Table 1. Statistics of the difference in SSA retrievals from almucantar scans of V2 and V3 for the Mezaira site. The difference is defined as V3-V2. The numbers in parentheses are standard deviations.









| AOD(440 nm) | VMR fine | STD fine | VMR coarse | STD coarse | Number of retrievals |
|---|---|---|---|---|---|
| < 0.2 | -0.008 (0.010) | -0.0024 (0.028) | -0.003 (0.119) | -0.010 (0.024) | 1261 |
| > 0.2 | -0.007 (0.011) | 0.000 (0.028) | 0.020 (0.087) | -0.003 (0.021) | 3848 |


Table 2. Statistics of the difference in volume median radius (VMR) in microns and width of particle size distribution (STD) retrievals of V3 and for the Mezaira site for almucantar retrievals with AOD (440)>0.2. The difference is defined as V3 -V2. The numbers in parentheses are standard deviations.










| AOD(440 nm) | RRI (440 nm) | RRI (675 nm) | RRI (870 nm) | RRI (1020 nm) | Number of retrievals |
|---|---|---|---|---|---|
| > 0.4 | 0.007 (0.031) | 0.010 (0.019) | 0.011 (0.015) | 0.018 (0.016) | 1473 |
| 0.4-0.6 | 0.005 (0.033) | 0.011 (0.021) | 0.012 (0.016) | 0.021 (0.017) | 1012 |
| > 0.6 | 0.009 (0.023) | 0.0085 (0.014) | 0.007 (0.013) | 0.013 (0.013) | 461 |

Table 3. Statistics of the difference in the real part of refractive index (RRI) retrievals of V2 and V3 for the Mezaira site. The

difference is defined as V3-V2. The numbers in parentheses are standard deviations.









| AOD(440 nm) | SSA (440 nm) | SSA (675 nm) | SSA (870 nm) | SSA (1020 nm) | Number of retrieval |
|---|---|---|---|---|---|
| > 0.4 | 0.002 (0.007 ) | 0.001 (0.008) | -0.001 (0.010) | -0.001 (0.012) | 709 |
| 0.4-0.6 | 0.001 (0.008) | 0.000 (0.009) | -0.002 (0.011) | -0.002 (0.013) | 404 |
| > 0.6 | 0.002 (0.006) | 0.001 (0.007) | 0.001 (0.007) | 0.001 (0.011) | 305 |

Table 4. Statistics of the difference in SSA retrievals of V2 and V3 for GSFC site. The difference is defined as V3-V2. The numbers in parentheses are standard deviations.









| AOD(440 nm) | VMR fine | STD fine | VMR coarse | STD coarse | Number of retrieval |
|---|---|---|---|---|---|
| < 0.2 | 0.002 (0.012) | 0.018 (0.031) | 0.027 (0.173) | -0.013 (0.034) | 6016 |
| > 0.2 | 0.006 (0.008) | 0.001 (0.022) | 0.063 (0.172) | -0.009 (0.030) | 1921 |


Table 5. Statistics of the difference in volume median radius (VMR) and width of particle size distribution (STD) retrievals of V2 and V3 for GSFC site. The difference is defined as V3 -V2. The numbers in parentheses are standard deviations.










| AOD(440 nm) | RRI (440 nm) | RRI (675 nm) | RRI (870 nm) | RRI (1020 nm) | Number of retrieval |
|---|---|---|---|---|---|
| > 0.4 | 0.032 (0.020) | 0.0182 (0.014) | 0.0147 (0.013) | 0.015 (0.012) | 709 |
| 0.4-0.6 | 0.038 (0.021) | 0.023 (0.015) | 0.018 (0.013) | 0.018 (0.012) | 404 |
| > 0.6 | 0.025 (0.017) | 0.013 (0.012) | 0.010 (0.011) | 0.011 (0.011) | 305 |

Table 6. Statistics of the difference in the real part of refractive index (RRI) retrievals of V2 and V3 for GSFC site. The difference is defined as V3 – V2. The numbers in parentheses are standard deviations.








| AOD(440 nm) | SSA (440 nm) | SSA (675 nm) | SSA (870 nm) | SSA (1020 nm) | Number of retrieval |
|---|---|---|---|---|---|
| > 0.4 | 0.005 (0.006) | 0.010 (0.008) | 0.001 (0.010) | -0.004 (0.013) | 1572 |
| 0.4-0.6 | 0.006 (0.007) | 0.011 (0.009) | 0.002 (0.012) | -0.002 (0.015) | 790 |
| > 0.6 | 0.005 (0.005) | 0.009 (0.007) | -0.001 (0.008) | -0.007 (0.011) | 782 |

Table 7. Statistics of the difference in SSA retrievals of V2 and V3 for Mongu site. The difference is defined as V3-V2. The numbers in parentheses are standard deviations.





| AOD(440 nm) | VMR fine | STD fine | VMR coarse | STD coarse | Number of retrieval |
|---|---|---|---|---|---|
| < 0.2 | 0.0084 (0.011) | 0.018 (0.043) | 0.128 (0.209) | -0.005 (0.038) | 1906 |
| > 0.2 | 0.001 (0.004) | -0.002 (0.019) | 0.129 (0.199) | -0.011 (0.031) | 2755 |

Table 8. Statistics of the difference in volume median radius (VMR) and width of particle size distribution (STD) retrievals of V2 and V3 for Mongu site. The difference is defined as V3 -V2. The numbers in parentheses are standard deviations.








| AOD(440 nm) | RRI (440 nm) | RRI (675 nm) | RRI (870 nm) | RRI (1020 nm) | Number of retrieval |
|---|---|---|---|---|---|
| > 0.4 | 0.019 (0.025) | 0.020 (0.020) | 0.019 (0.018) | 0.022 (0.017) | 1572 |
| 0.4-0.6 | 0.020 (0.029) | 0.020 (0.023) | 0.019 (0.020) | 0.022 (0.019) | 790 |
| > 0.6 | 0.0191 (0.020) | 0.020 (0.016) | 0.019 (0.015) | 0.022 (0.016) | 782 |

Table 9. Statistics of the difference in the real part of refractive index (RRI) retrievals of V2 and three for Mongu site. The difference is defined as V3-V2. The numbers in parentheses are standard deviations.










| AOD(440 nm) | SSA (440 nm) | SSA (675 nm) | SSA (870 nm) | SSA (1020 nm) | Number of retrieval |
|---|---|---|---|---|---|
| > 0.4 | 0.005 (0.008) | 0.011 (0.008) | -0.001 (0.009) | -0.002 (0.011) | 5667 |
| 0.4-0.6 | 0.005 (0.009) | 0.012 (0.009) | -0.000 (0.011) | -0.000 (0.011) | 1907 |
| > 0.6 | 0.005 (0.007) | 0.011 (0.007) | -0.002 (0.009) | -0.002 (0.010) | 3760 |

Table 10. Statistics of the difference in SSA retrievals of V2 and three for Kanpur site. The difference is defined as V3-V2. The numbers in parentheses are standard deviations.










| AOD(440 nm) | VMR fine | STD fine | VMR coarse | STD coarse | Number of retrieval |
|---|---|---|---|---|---|
| < 0.2 | -0.001 (0.010) | 0.020 (0.024) | 0.106 (0.106) | 0.004 (0.0206) | 68 |
| > 0.2 | -0.002 (0.011) | 0.007 (0.033) | 0.135 (0.138) | 0.016 (0.027) | 6517 |

Table 11. Statistics of the difference in volume median radius (VMR) and width of particle size distribution (STD) retrievals
of V2 and three for Kanpur site. The difference is defined as V3 -V2. The numbers in parentheses are standard deviations.









| AOD(440 nm) | RRI (440 nm) | RRI (675 nm) | RRI (870 nm) | RRI (1020 nm) | Number of retrieval |
| --- | --- | --- | --- | --- | --- |
| > 0.4 | 0.020 (0.039) | 0.014 (0.028) | 0.007 (0.023) | 0.007 (0.022) | 5667 |
| 0.4-0.6 | 0.024 (0.044) | 0.018 (0.030) | 0.0103 (0.025) | 0.010 (0.024) | 1907 |
| > 0.6 | 0.018 (0.037) | 0.012 (0.026) | 0.006 (0.022) | 0.006 (0.021) | 3760 |


Table 12. Statistics of the difference in the real part of refractive index (RRI) retrievals of V2 and three for Kanpur site. The difference is defined as V3-V2. The numbers in parentheses are standard deviations.










| Scattering Angle (°) | Az (30° SZA) | VZA (30° SZA) | Az (60° SZA) | VZA (60° SZA) | Az (75° SZA) | VZA 75° SZA) |
|---|---|---|---|---|---|---|
| 3 | 5.98364 | 30.13573 | 3.463048 | 60.04532 | 3.105854 | 75 |
| 3.5 | 6.974072 | 30.18459 | 4.039779 | 60.06168 | 3.623507 | 75 |
| 4 | 7.961385 | 30.24087 | 4.616306 | 60.08055 | 4.141165 | 75 |
| 5 | 9.924985 | 30.37551 | 5.768632 | 60.1258 | 5.176499 | 75 |
| 6 | 11.87122 | 30.53926 | 6.919798 | 60.18105 | 6.211861 | 75 |
| 6 | 11.87122 | 30.53926 | 6.919798 | 60.18105 | 6.211861 | 75 |
| 7 | 13.79706 | 30.73164 | 8.069578 | 60.24627 | 7.247257 | 75 |
| 8 | 15.6997 | 30.95213 | 9.217749 | 60.32141 | 8.282693 | 75 |
| 10 | 19.4254 | 31.47495 | 11.50839 | 60.5013 | 10.35371 | 75 |
| 12 | 23.03098 | 32.10229 | 13.79004 | 60.72028 | 12.42495 | 75 |
| 14 | 26.50341 | 32.82811 | 16.06107 | 60.97784 | 14.49647 | 75 |
| 16 | 29.83387 | 33.64596 | 18.31997 | 61.27339 | 16.56832 | 75 |
| 18 | 33.0174 | 34.54924 | 20.56534 | 61.60625 | 18.64053 | 75 |
| 20 | 36.05239 | 35.53135 | 22.79588 | 61.97568 | 20.71317 | 75 |
| 25 | 43.00307 | 38.2899 | 28.30005 | 63.05379 | 25.89696 | 75 |
| 30 | 49.10661 | 41.40962 | 33.69007 | 64.34109 | 31.08454 | 75 |
| 35 | 54.47036 | 44.81336 | 38.95658 | 65.8218 | 36.27682 | 75 |
| 40 | 59.21027 | 48.43924 | 44.09531 | 67.47899 | 41.47479 | 75 |
| 45 | 63.43495 | 52.23876 | 49.10661 | 69.29519 | 46.67956 | 75 |
| 50 | 67.23952 | 56.17416 | 53.99479 | 71.25276 | 51.89235 | 75 |
| 60 | 73.89789 | 64.34109 | 63.7035 | 75 | 62.3479 | 75 |
| 70 | 79.6859 | 72.7706 | 75.27607 | 75 | 72.8557 | 75 |
| 80 | 96.00147 | 75 | 86.96853 | 75 | 83.43567 | 75 |
| 90 | 117.6521 | 75 | 98.89943 | 75 | 94.11719 | 75 |
| 100 | 145.4518 | 75 | 111.2406 | 75 | 104.9462 | 75 |
| 120 | N/A | N/A | 138.8002 | 75 | 127.423 | 75 |
| 140 | N/A | N/A | N/A | N/A | 153.2324 | 75 |
| 160 | N/A | N/A | N/A | N/A | N/A | N/A |
| 180 | N/A | N/A | N/A | N/A | N/A | N/A |

Table 13. Hybrid sky scan with azimuth (AZ) and view zenith angle (VZA) for three selected solar zenith angles at 30°, 60° and 75°.






| AOD (440) | 0.03 | 0.07 | 0.1 | 0.2 | 0.3 | 0.4 | 0.5 | 0.6 | 0.7 |
|---|---|---|---|---|---|---|---|---|---|
| U27, 440 | 0.0872 | 0.064 | 0.052 | 0.037 | 0.028 | 0.023 | 0.019 | 0.017 | 0.017 |
| U27, 675 | 0.089 | 0.068 | 0.057 | 0.041 | 0.034 | 0.029 | 0.023 | 0.021 | 0.021 |
| U27, 870 | 0.103 | 0.081 | 0.069 | 0.052 | 0.043 | 0.037 | 0.029 | 0.026 | 0.027 |
| U27, 1020 | 0.110 | 0.088 | 0.075 | 0.057 | 0.048 | 0.043 | 0.033 | 0.030 | 0.031 |

Table 14. Average SSA uncertainties estimated at GSFC.








| AOD(440) | 0.03 | 0.07 | 0.1 | 0.2 | 0.3 | 0.4 | 0.5 | 0.6 | 0.7 |
|---|---|---|---|---|---|---|---|---|---|
| U27, 440 | 0.064 | 0.064 | 0.061 | 0.046 | 0.035 | 0.028 | 0.023 | 0.020 | 0.017 |
| U27, 675 | 0.053 | 0.049 | 0.046 | 0.035 | 0.028 | 0.022 | 0.018 | 0.015 | 0.013 |
| U27, 870 | 0.064 | 0.055 | 0.049 | 0.037 | 0.029 | 0.022 | 0.018 | 0.015 | 0.013 |
| U27, 1020 | 0.081 | 0.066 | 0.056 | 0.039 | 0.031 | 0.024 | 0.019 | 0.016 | 0.014 |

Table 15. Average SSA uncertainties estimated at Mezaira.









| AOD(440) | 0.03 | 0.07 | 0.1 | 0.2 | 0.3 | 0.4 | 0.5 | 0.6 | 0.7 |
|---|---|---|---|---|---|---|---|---|---|
| U27, 440 | 0.078 | 0.064 | 0.057 | 0.044 | 0.036 | 0.029 | 0.025 | 0.023 | 0.021 |
| U27, 675 | 0.077 | 0.066 | 0.059 | 0.045 | 0.037 | 0.032 | 0.028 | 0.026 | 0.025 |
| U27, 870 | 0.089 | 0.078 | 0.070 | 0.053 | 0.042 | 0.036 | 0.032 | 0.030 | 0.028 |
| U27, 1020 | 0.094 | 0.084 | 0.076 | 0.058 | 0.045 | 0.040 | 0.034 | 0.032 | 0.030 |

Table 16. Average SSA uncertainties estimated at Mongu.









| AOD(440) | 0.03 | 0.07 | 0.1 | 0.2 | 0.3 | 0.4 | 0.5 | 0.6 | 0.7 |
|---|---|---|---|---|---|---|---|---|---|
| U27, 440 | 0.091 | 0.078 | 0.070 | 0.048 | 0.036 | 0.030 | 0.025 | 0.021 | 0.019 |
| U27, 675 | 0.073 | 0.064 | 0.058 | 0.041 | 0.031 | 0.028 | 0.024 | 0.022 | 0.021 |
| U27, 870 | 0.078 | 0.069 | 0.062 | 0.045 | 0.034 | 0.029 | 0.026 | 0.024 | 0.023 |
| U27, 1020 | 0.079 | 0.070 | 0.063 | 0.046 | 0.035 | 0.030 | 0.027 | 0.026 | 0.025 |

Table 17. Average SSA uncertainties estimated at Kanpur.









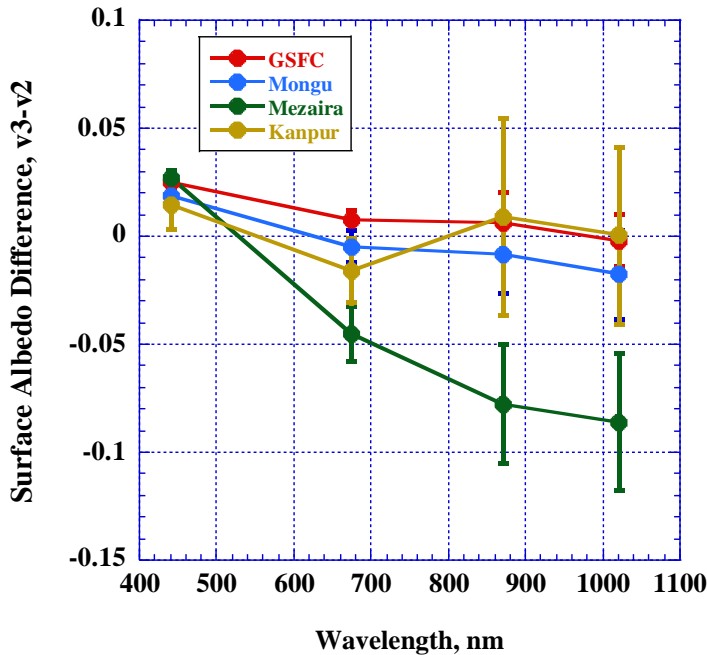

Figure 1. The V2 and V3 average differences and standard deviations in surface albedo calculated using BRDF parameters at four sites: Goddard Space Flight Center (GSFC, USA), Mezaira (UAE), Mongu (Zambia), and Kanpur (India).







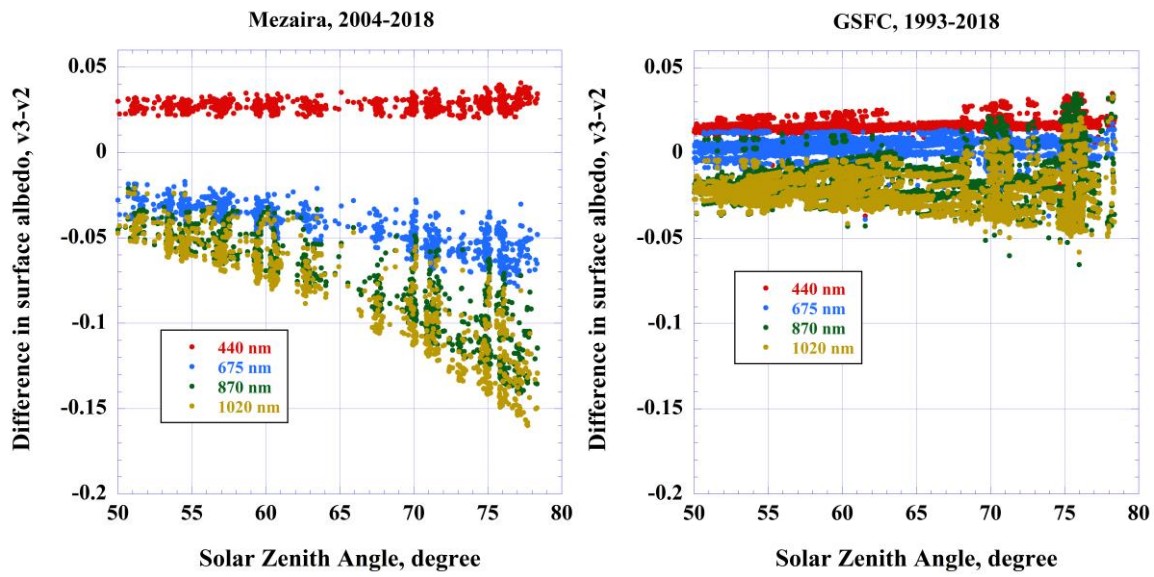

Figure 2. Difference in surface albedo calculated with BRDF from V2 and V3 for a) Mezaira (2004-2018) and b) GSFC (1993-2018).









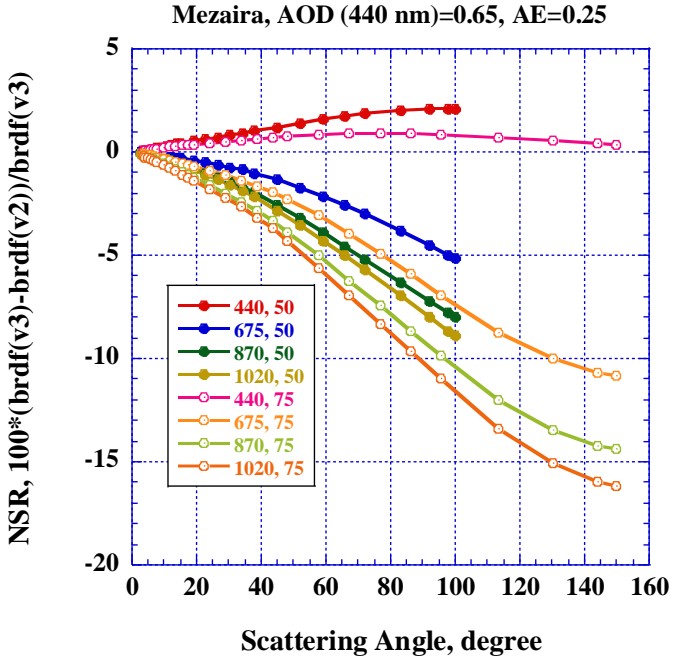

Figure 3. The relative percent difference in normalized sky radiances (NSR) calculated with BRDF parameters from V2 and
V3 for the Mezaira site.








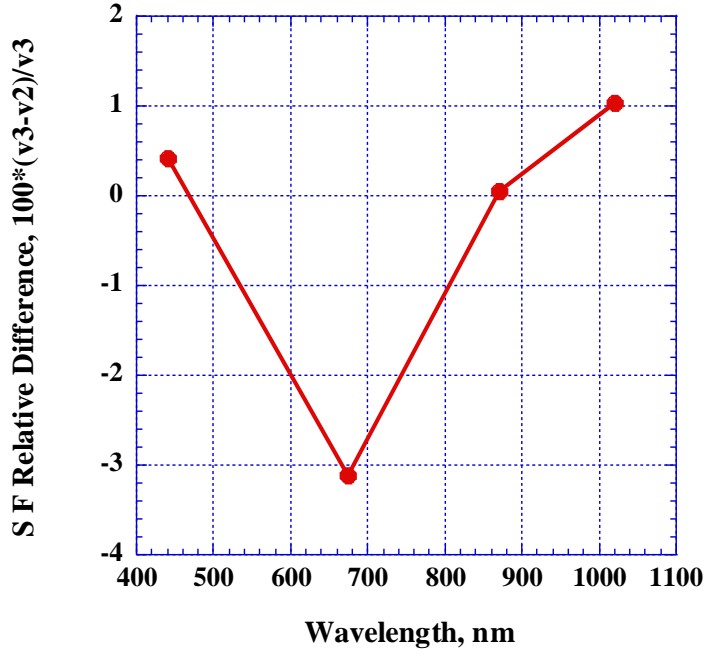

Figure 4. The relative percent difference in spectral solar flux data used in V2 (ASTM E-490 https://www.nrel.gov/grid/solar-resource/spectra.html) and V3 (Coddington et al., 2012) for four standard AERONET sky wavelengths (i. e. 440, 675, 870, and 1020 nm).





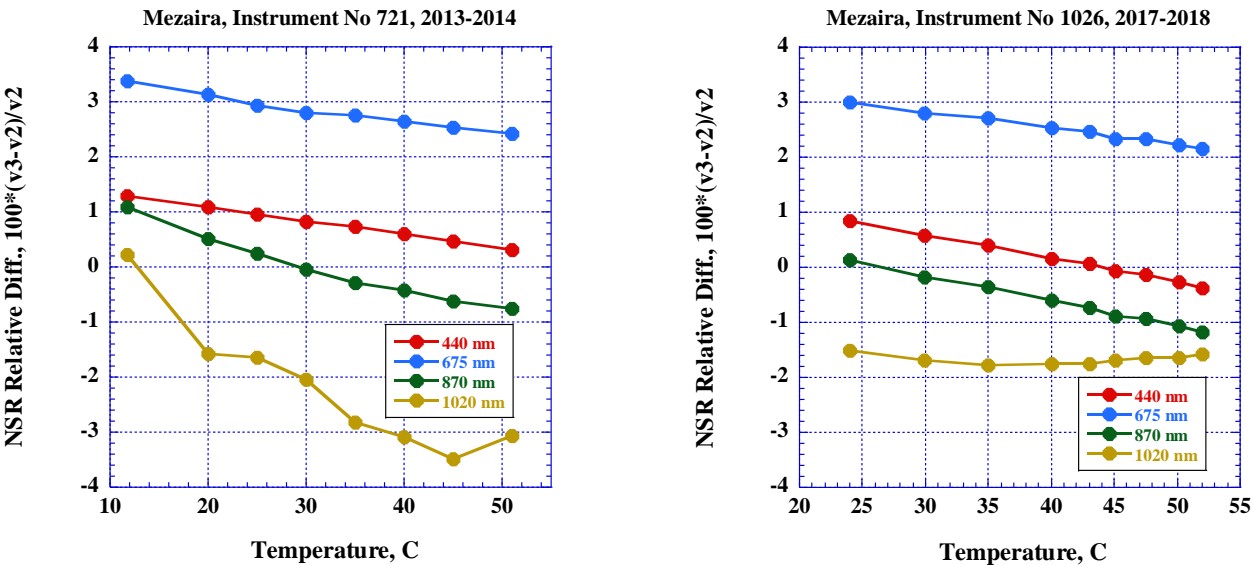

Figure 5. Effect of solar spectrum and temperature correction on relative percent difference in normalized sky radiances (NSR) for Mezaira a) instrument 721 b) instrument 1026.





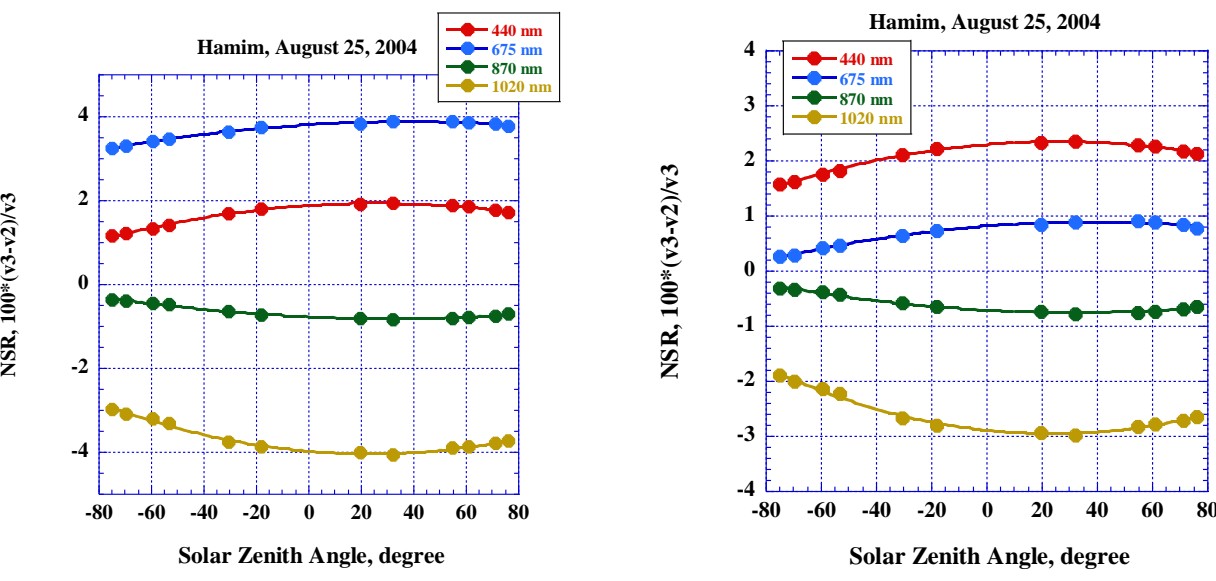


Figure 6. Effect of solar flux and temperature correction on relative percent difference in normalized sky radiances for Hamim:
a) joint effect of temperature correction and solar flux b) the effect of solar flux is subtracted.







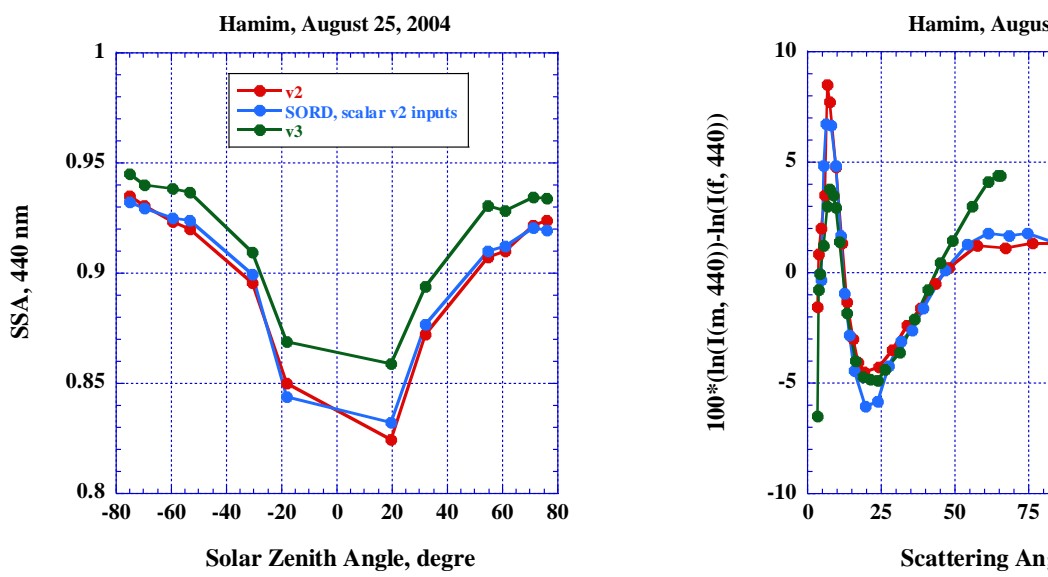

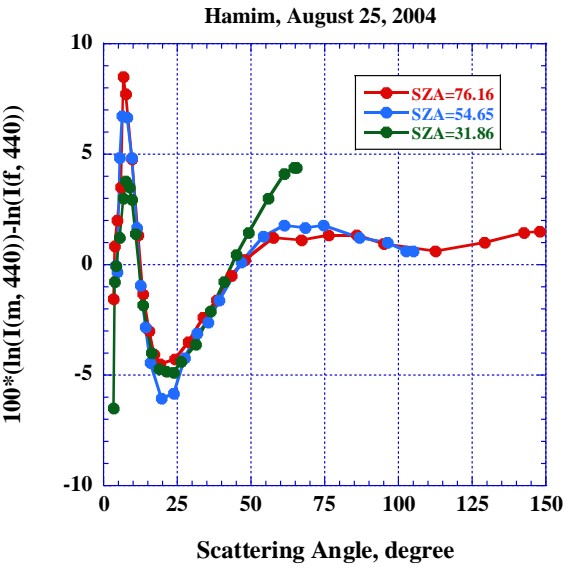

Figure 7. Data from Hamim (UAE) on 25 August 2004 showing a) diurnal variability of SSA at 440 nm retrieved from almucantars by the V2 and V3 algorithms and algorithm of V3 in scalar mode using inputs of V2 b) the relative percent
difference between measured and fitted NSR at 440 nm for three different SZA.








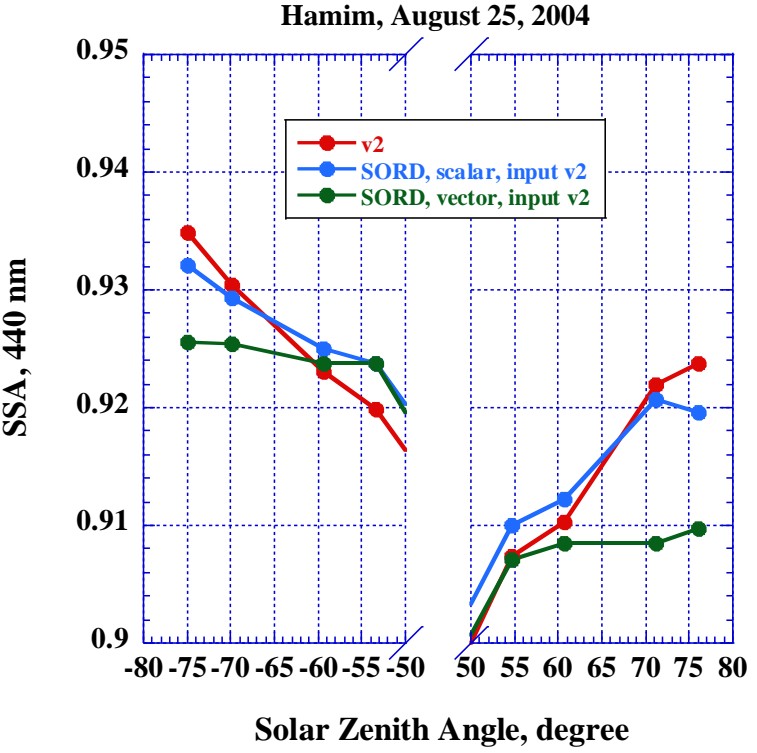

Figure 8. Hamim on 25 August 2004. The diurnal variability of Level 2 SSA retrievals at 440 nm for three combinations of SORD and inversion inputs: V2 (red), SORD scalar mode (blue), and SORD vector mode using V2 inputs (green).





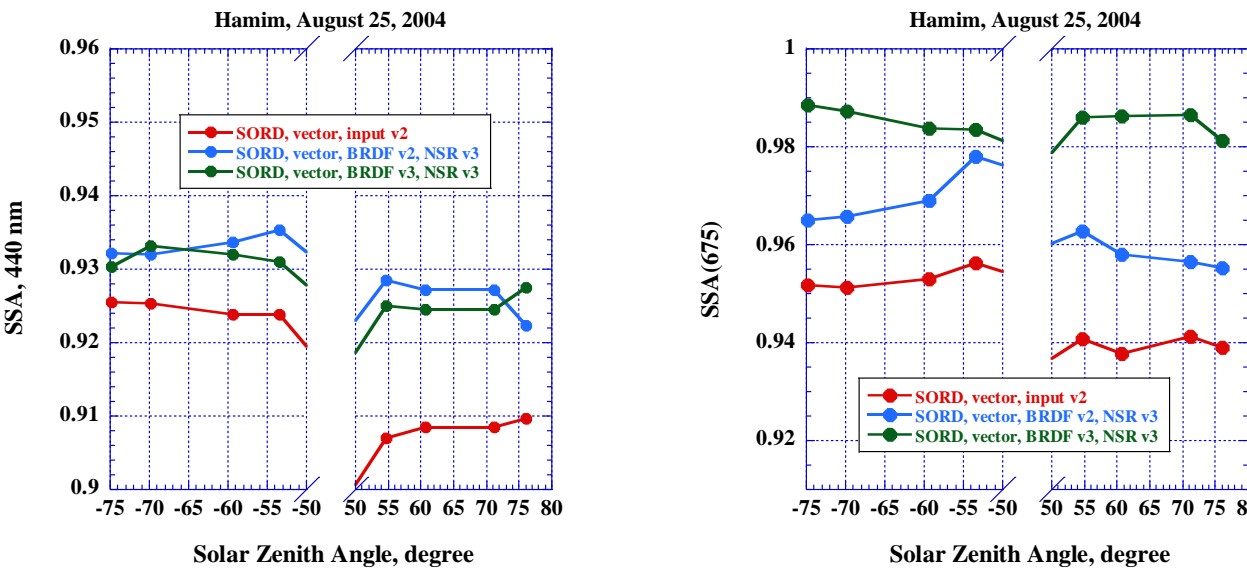

Figure 9. Hamim on 25 August 2004. The diurnal variability of Level 2 SSA retrievals at 440 nm for three combinations of SORD and inversion inputs: V2 inputs (red), combined inputs with BRDF parameters of V2 and NSR of V3 (blue), and combined inputs with BRDF parameters of V3 and NSR of V3 (green) for a) 440 nm and b) 675 nm.





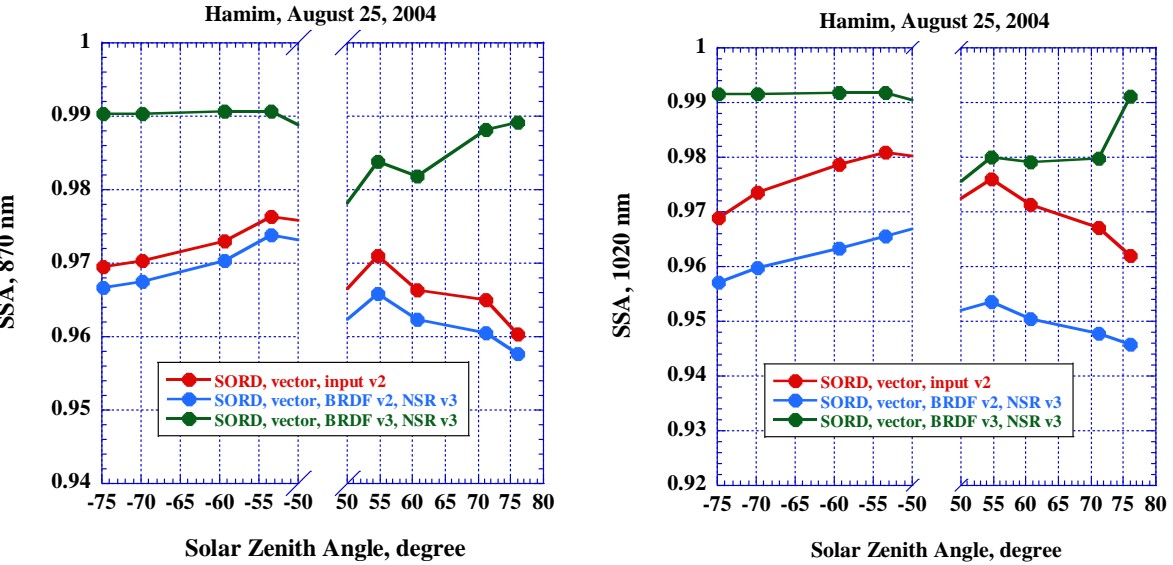

Figure 10. The same as Figure 9 but for a) 870 nm and b) 1020 nm.









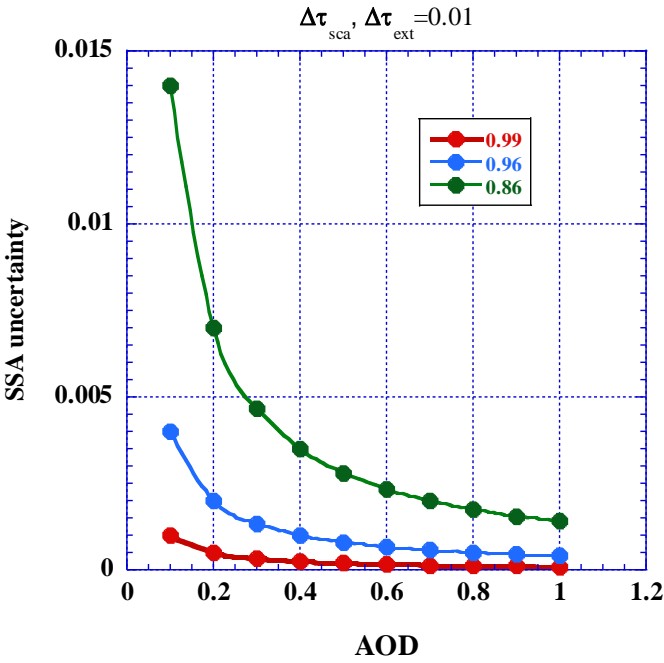

Figure 11. The uncertainty in SSA simulated using Eq. (1) for the three values of SSA: 0.99 (red), 0.96 (blue) and 0.86 (green).









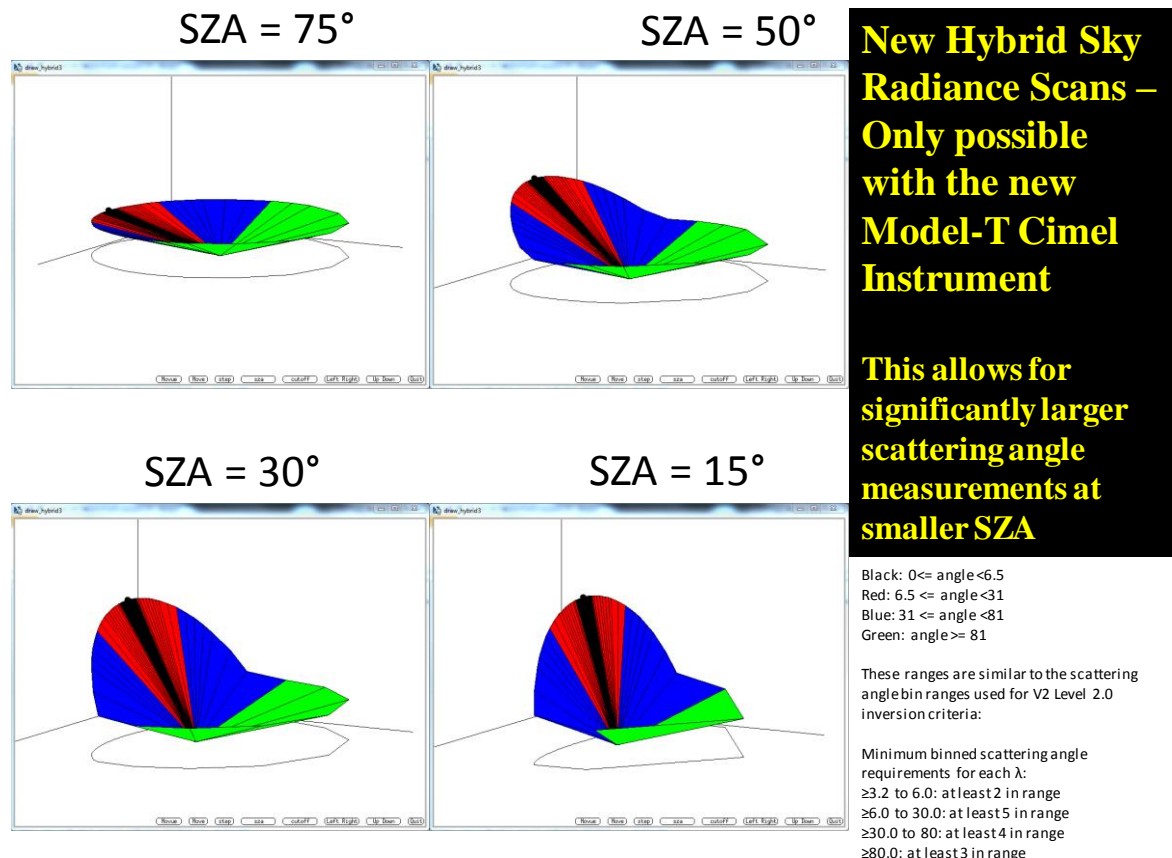

Figure 12. Hybrid scan geometry depicted for four values of SZA. See explanations in text.








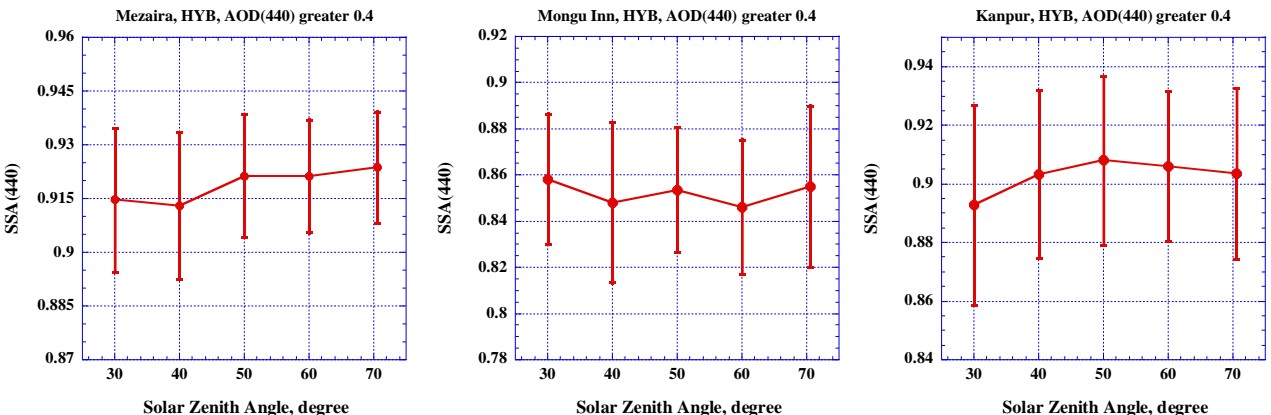

Figure 13. Dependence of 440 nm SSA hybrid scan retrievals on SZA for three AERONET sites: a) Mezaira, b) Mongu Inn, c) Kanpur.










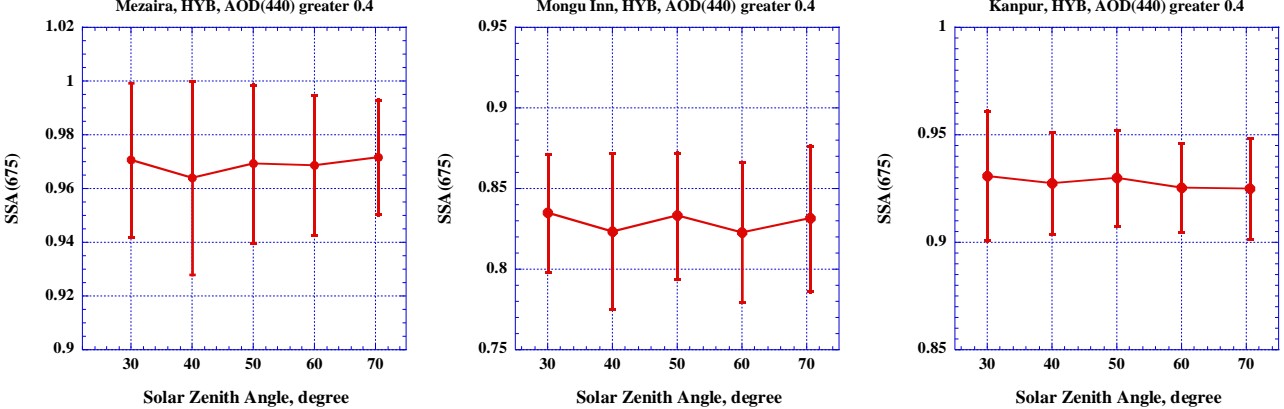

Figure 14. Dependence of 675 nm SSA hybrid scan retrievals on SZA for three AERONET sites: a) Mezaira, b) Mongu Inn, c) Kanpur.










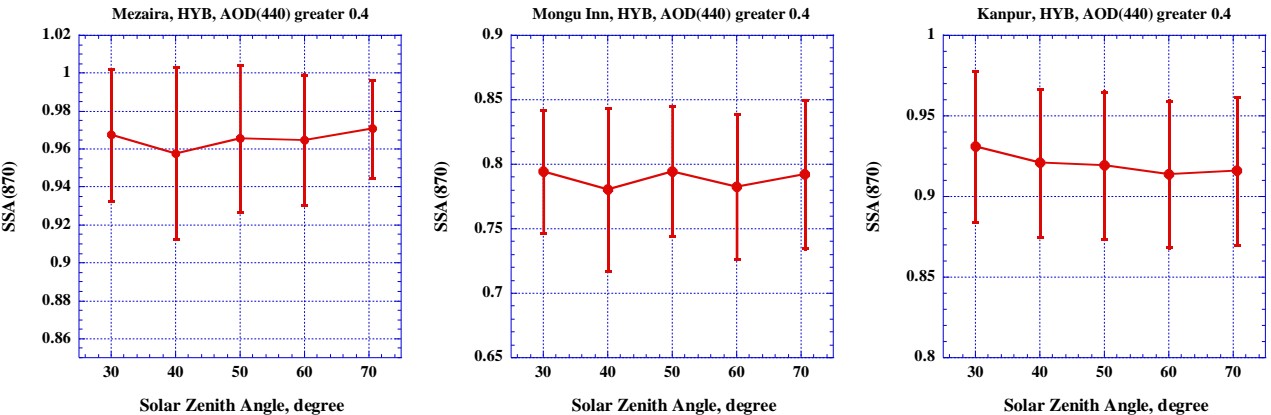

Figure 15. Dependence of 870 nm SSA hybrid scan retrievals on SZA for three AERONET sites: a) Mezaira, b) Mongu Inn,
c) Kanpur.





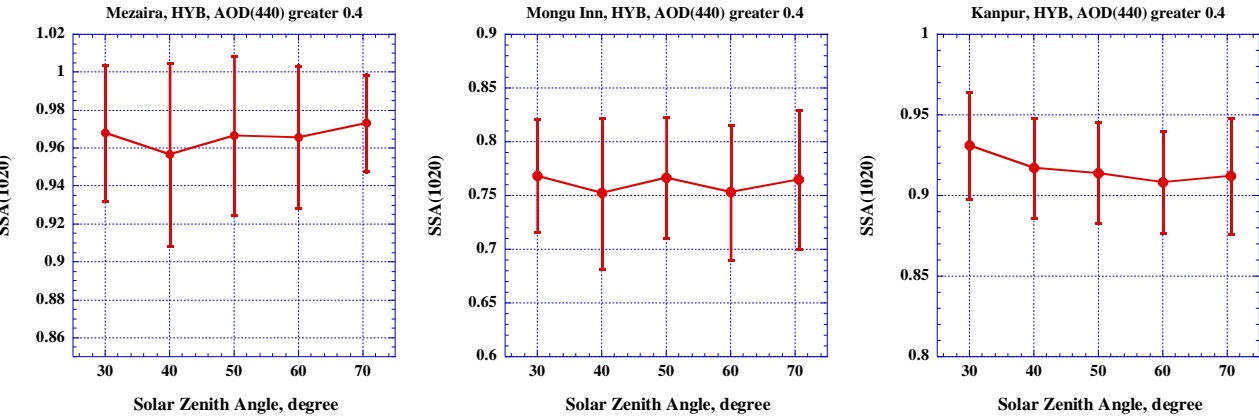


Figure 16. Dependence of 1020 nm SSA hybrid scan retrievals on SZA for three AERONET sites: a) Mezaira, b) Mongu Inn, c) Kanpur.








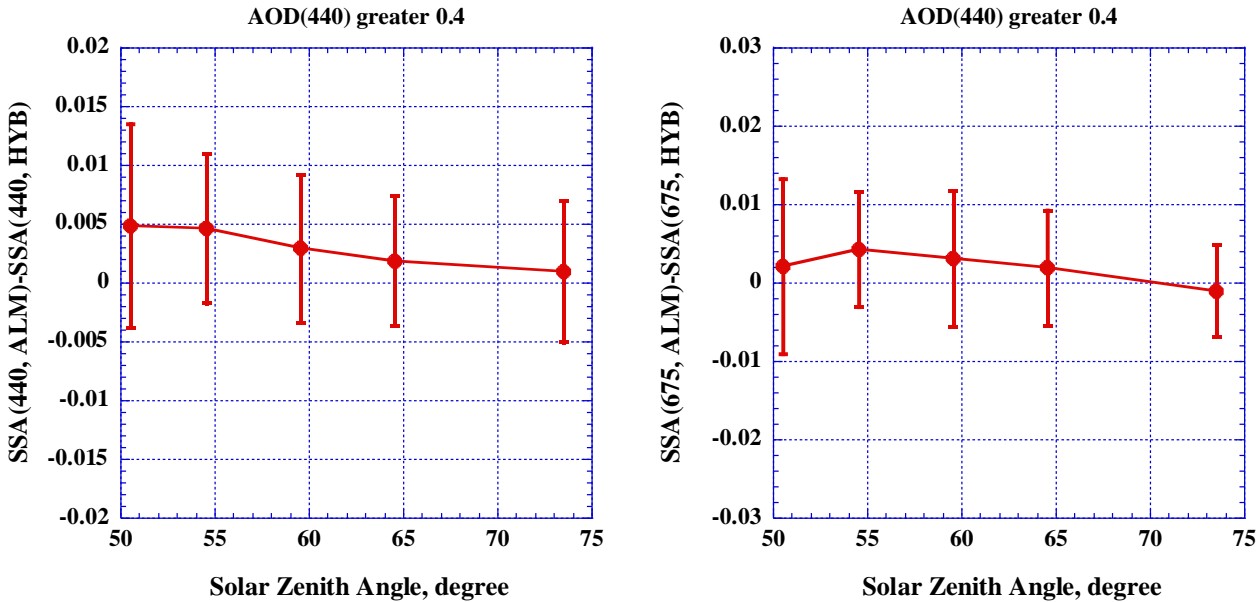

Figure 17. Difference in SSA retrieved from HYB and ALM scans. a) 440 nm, b) 675 nm.







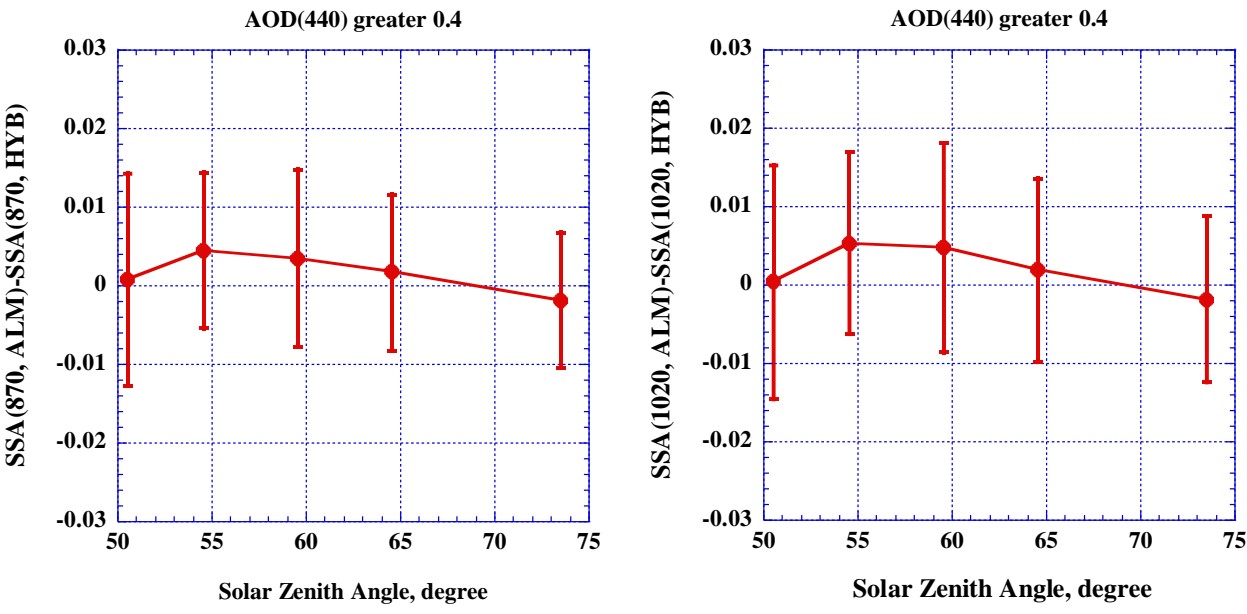

Figure 18. Difference in SSA retrieved from HYB and ALM scans. a) 870 nm, b) 1020 nm.








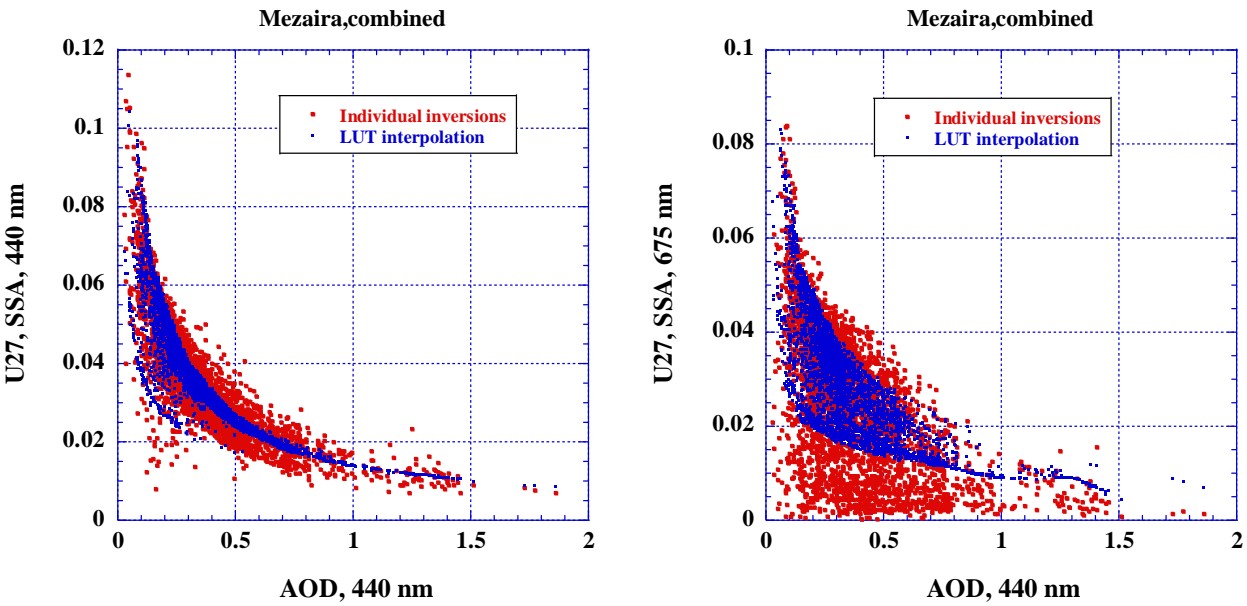

Figure 19. Comparison of SSA U27 estimated for individual inversions and by interpolation over LUT for Mezaira site: a) 440 nm, b) 675 nm.



Figure 20. Uncertainties for SSA retrievals at GSFC site, a) 440 nm, b) 675 nm, c) 870 nm, d)1020 nm.



Figure 21. Uncertainties for SSA retrievals at Mezaira site, a) 440 nm, b) 675 nm, c) 870 nm, d)1020 nm.




Figure 22. SSA uncertainties generated by averaging over AOD bins with subsequent interpolation to the regularly spaced
AOD grid; a) GSFC, b) Mezaira, c) Mongu, d) Kanpur.




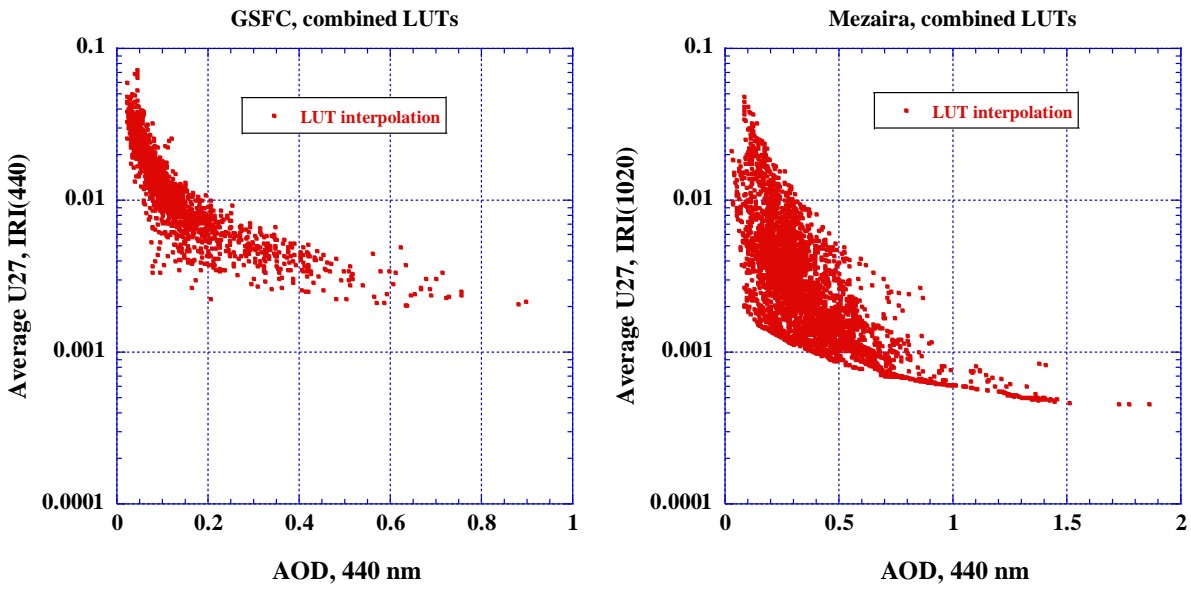


Figure 23 Uncertainties of the imaginary part of refractive index estimates, a) GSFC, b) Mezaira.









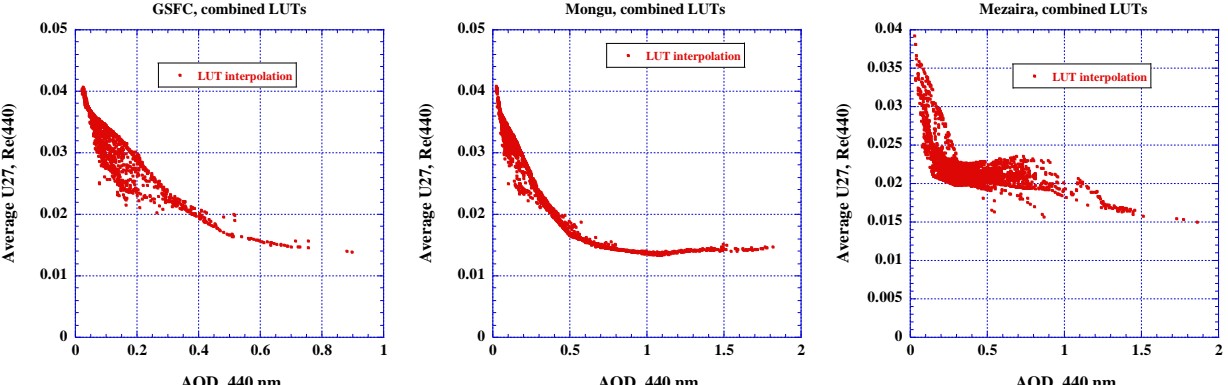

Figure 24. Uncertainties of the real part of refractive index estimated at 440 nm, a) GSFC, b) Mongu, c) Mezaira.










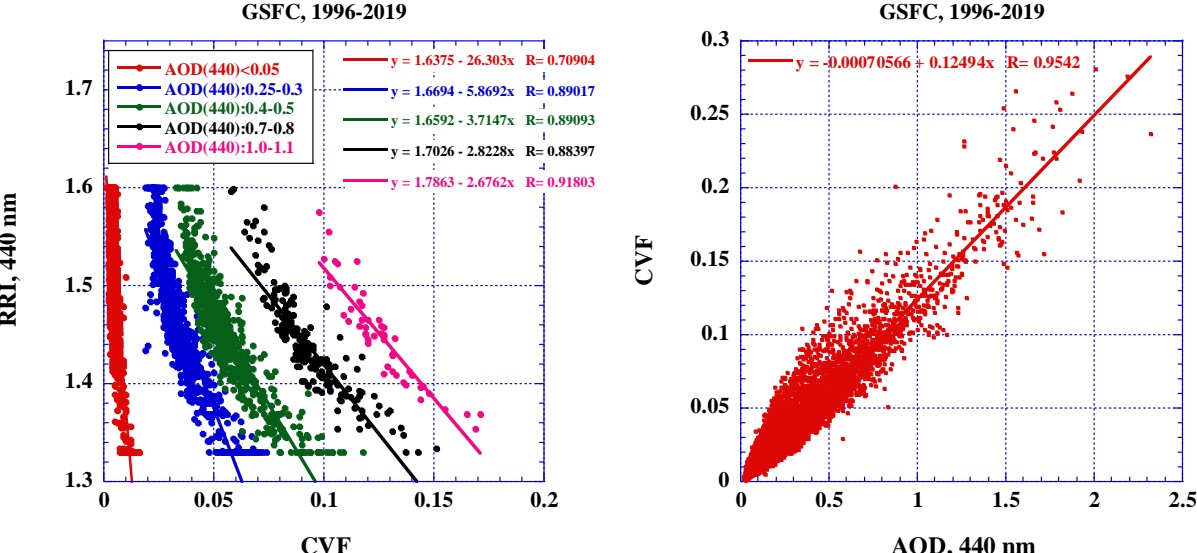

Figure 25. Analysis of correlation between the real part of refractive index and fine mode concentration at GSFC site.










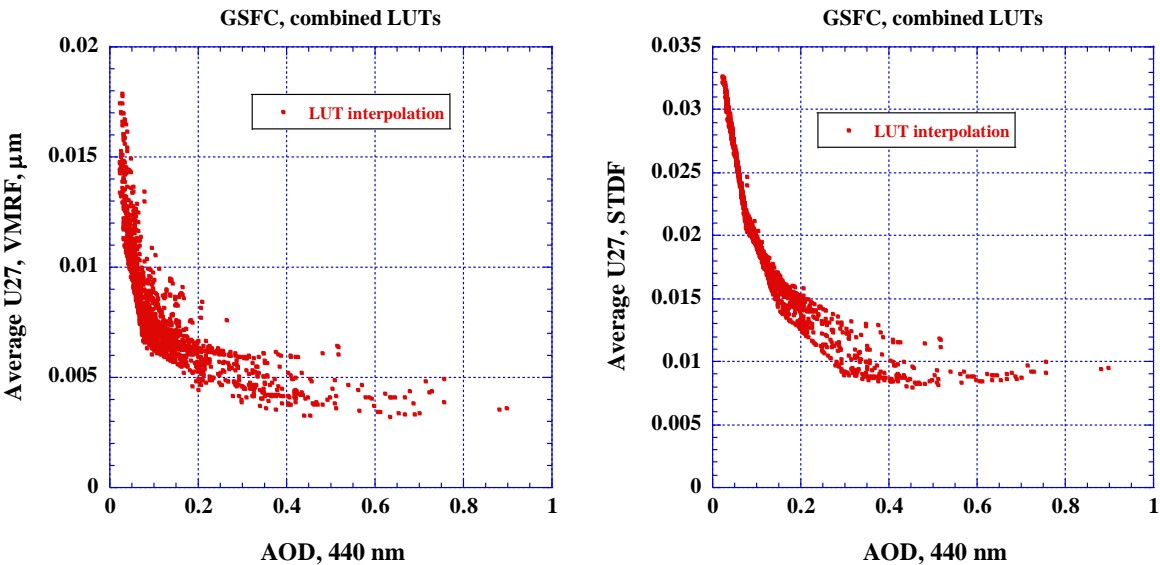

Figure 26. Uncertainties for a) volume median radius of fine mode and b) width of size distribution for fine mode estimated at GSFC site.









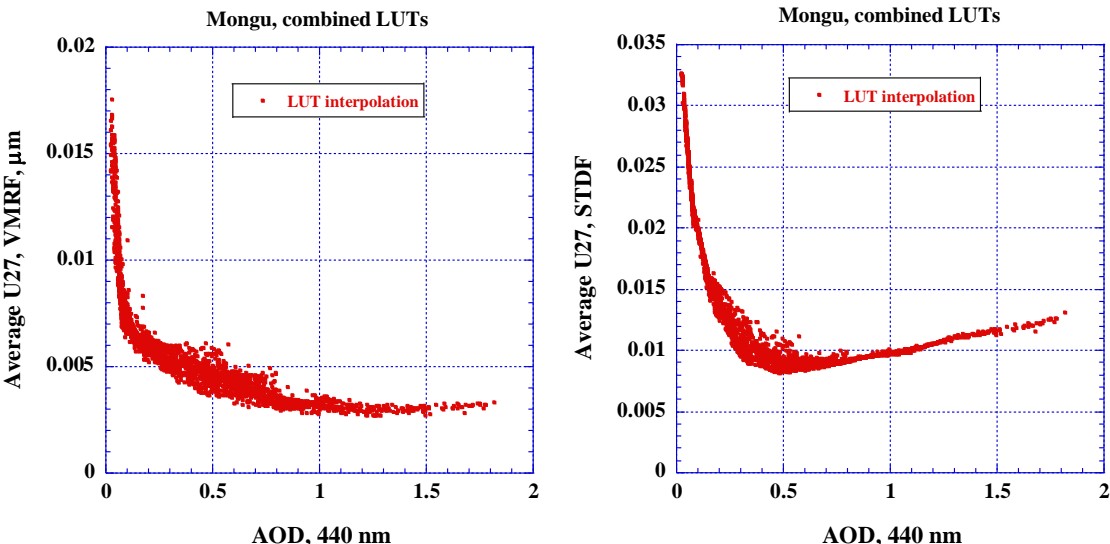

Figure 27. Uncertainties for a) volume median radius of fine mode and b) width of size distribution for fine mode estimated at Mongu site.









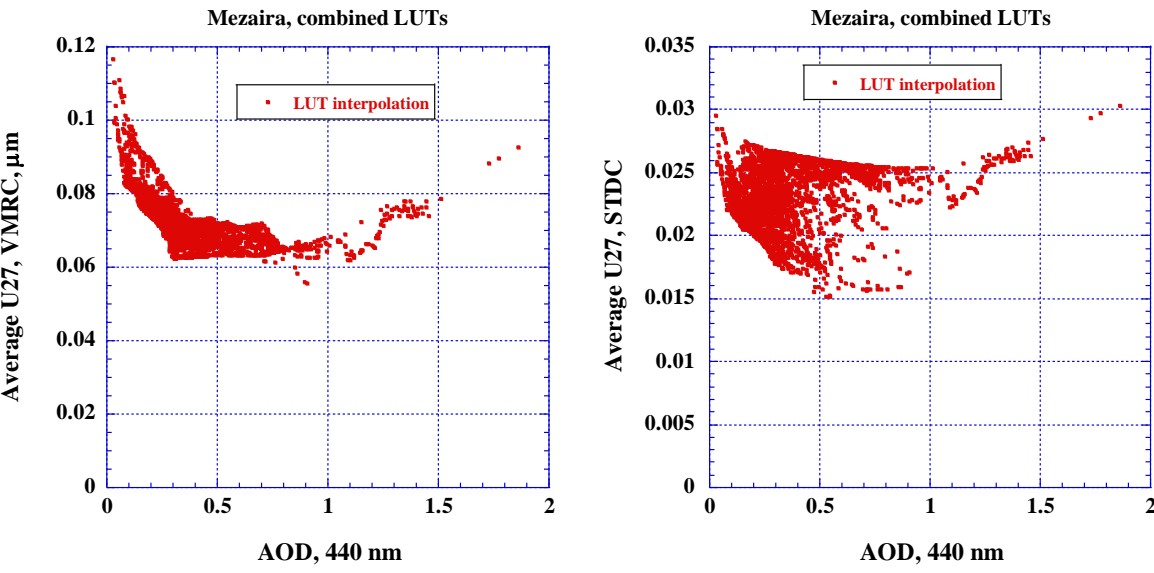

Figure 28. Uncertainties for a) volume median radius of coarse mode and b) width of size distribution for coarse mode estimated at Mezaira site.









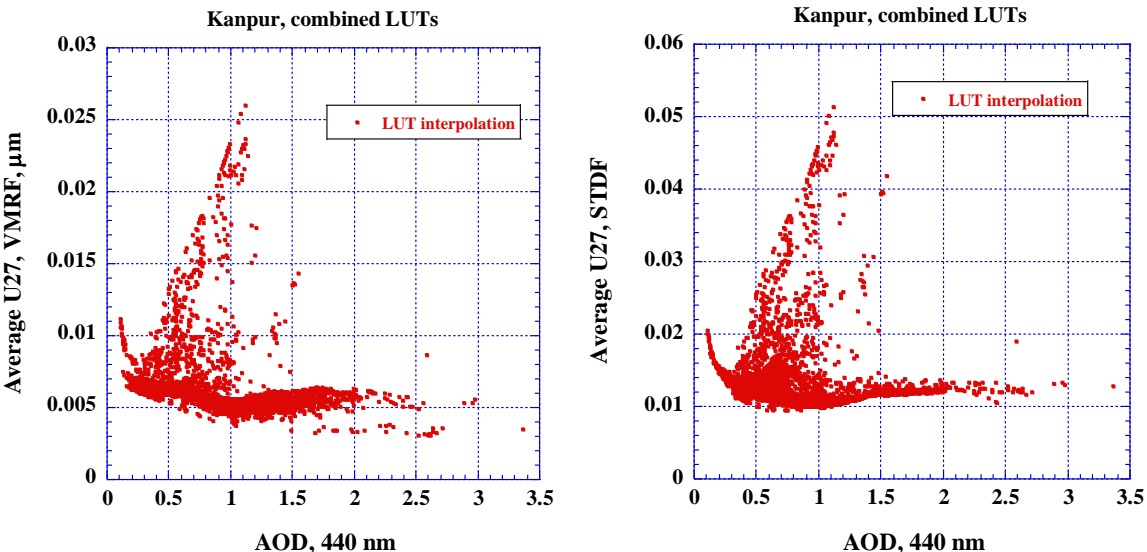

Figure 29. Uncertainties for a) volume median radius of fine mode and b) width of size distribution for fine mode estimated at Kanpur site.







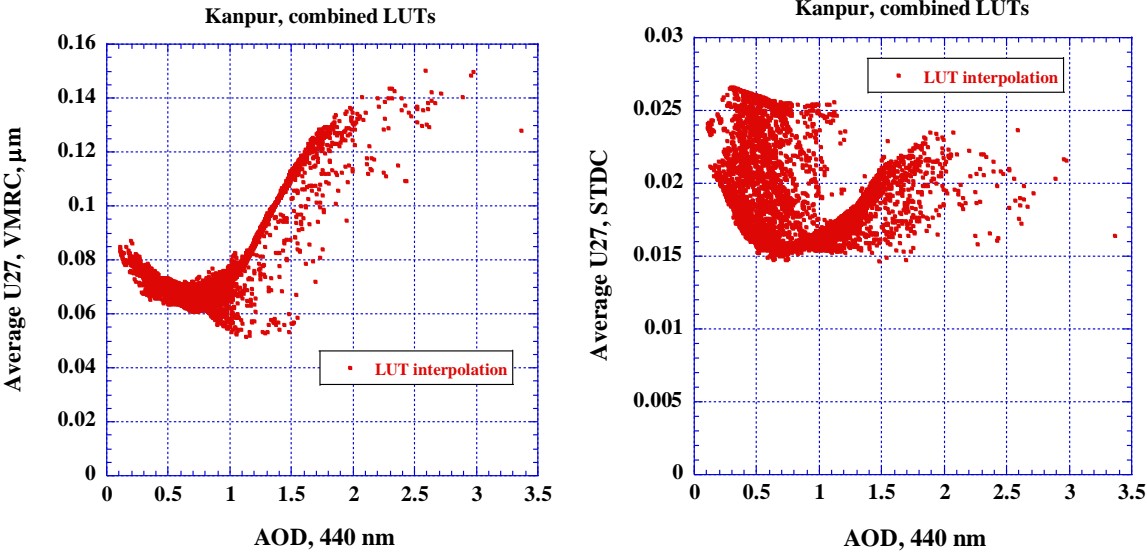

Figure 30. Uncertainties for a) volume median radius of coarse mode and b) width of size distribution for coarse mode
estimated at Kanpur site.
