# Peer review of "The AERONET Version 3 aerosol retrieval algorithm, associated uncertainties and comparisons to Version 2"

_Atmospheric Measurement Techniques, 2019_

## Referee Comment (RC1) · Anonymous Referee #2 · 10 Feb 2020

The present manuscript is devoted to describing the solution of two tasks: 1) The changes and additions to the V3 aerosol retrieval algorithm over that of V2 are presented and the potential effect of each change in V3 on aerosol retrievals is analyzed. Operational almucantar retrievals of V2 versus V3 were compared for four AERONET sites (GSFC, Mezaira, Mongu,and Kanpur).

2) A new approach to estimate uncertainties in the retrieved aerosol parameters was developed. The LUT approach was tested by generating U27 for aerosol retrievals at four selected AERONET sites representing differing aerosol types.

Besides these tasks, a new sky radiance angular distribution measurements scan,

called the hybrid, is introduced and discussed.

The topics in the manuscript are undoubtedly urgent. The problem is that certain questions within each of these tasks require a more detailed description than is done in this version of the manuscript. I think it is reasonable to submit a number of considered questions either after, or in parallel with, publishing this manuscript. It is hardly appropriate to expand substantially the given version of the manuscript. In particular, the manuscript already contains very many figures and tables, complicating the perception of the material. The second variant is to divide the text presented into two parts: description of the AERONET Version 3 and description of new approach to estimating the uncertainties in the retrieved aerosol parameters. In principle, these are two different tasks that can be described separately.

Major comments (in the order of their appearance in the text)

Line 223. While for the almucantar (ALM) observation geometry this is a reasonable assumption (e.g. Dubovik and King, 2000b; Torres et al., 2014), for other geometries the sensitivity to vertical structure of aerosol and gases in atmosphere can be important, especially at shorter wavelengths with relatively large Rayleigh scattering.

Please provide a reference or numerical estimates

Page 9 (3.2 Effects of changes in Extraterrestrial solar flux and temperature correction)

Figure 5 discusses the temperature dependences of NSR. Seemingly, the temperatures in the range of 10-50oC at the selected site are distributed non-uniformly, and points on the plots differ in statistical representativeness. Can this influence the result?

Section 4 (Hybrid scan: concept and retrieval scan).

From materials, now presented in section 4, we can gain only a general idea of the novelties associated with the new instrument type. If they are left in about the same form that we see in the manuscript, it is reasonable to consider questions, regarding the hybrid scan, in more detail and, presumably, in a separate publication. Maybe this

publication does already exist now?

Reading the section raises the following questions.

How many photometers, ensuring the new scanning geometry, have been installed and already operate now? Are they installed at all four sites, data from which are used in the present study, and how long ago? Does the aeronet.gsfc.nasa.gov site present information on the type of instrument (or instruments, if they operate in parallel) that is used at arbitrarily chosen observation site? In any case, I cannot see such data. Did the cloud screening procedure change after passing to new sensing geometry?

Results in subsection 4.2 are described very sparingly. Results in Figs. 17 and 18 can hardly be considered as an argument in favor of the good correspondence between results retrieved using two different scanning geometries, more so considering that the authors present no statistics that was used in these comparisons. For instance, it is unclear why the authors conclude that Đřt the same time the variability is increasing with increasing wavelength due to predominant contribution of fine mode aerosols to the generated statistics, and therefore much smaller AOD at the longest wavelengths which results in less sensitivity to aerosol absorption (line 496)?

Can you present data for at least a few observation sites that would show how much the number of retrievals increased after the new geometry was introduced?

Line 534. The radiometric calibration and solar spectrum irradiance uncertainties are combined in one bias because both of them affect the magnitude of sky radiances. What is the value of uncertainty, resulting from combining the radiometric calibration and solar spectrum irradiance uncertainties?

Subsection 5.4. In analyzing the results from retrieving aerosol characteristics, it is important for the reader to obtain information on what is the uncertainty degree of any characteristic at a specific site under certain atmospheric conditions. The authors carried out such an analysis for four sites (GSFC, Mezaira, Kanpur, and Mongu), which

correspond to four different aerosol types. It is reasonable to stress this in the text of the manuscript (put simply, starting an analysis, say, for GSFC, to prescribe aerosol type characteristic for this observation site in an explicit form). It would also be useful for the reader to see any climatologic data on characteristics for four selected sites (type of AOD distribution, relationship between fine and coarse fractions, characteristic types of the underlying surface, etc.) Then it will be easier for the reader to choose estimates of uncertainty that correspond to the specific data that he analyzes.

Line 666. Fig. 24 shows the uncertainties (by the U27 methodology) of the RRI at 440 nm estimated for the GSFC, Mongu, and Mezaira sites. Can any recommendations be obtained regarding RRI at other wavelengths?

Line 679. In contrast, the U27 estimated for SSA and observed variability of SSA retrievals are more consistent. Additionally, the upper and lower limit retrieval constraints of RRI (1.33 and 1.60, respectively) were reached at times, suggesting some potential instability in the retrievals of RRI. The percentage values of RRI retrievals reaching 1.33 and 1.6 (in parentheses) were estimated for all three sites for 440 nm AOD less than 0.2 and greater than 0.4. The estimates for smaller AOD bin show that RRI retrievals hit 1.6 boundary more often than that of 1.33: 1.6% (13.7%) for Mezaira, 3.1% (11.1%) for GSFC, and 2.2% (18.5%) for Mongu. The corresponding estimates for larger AOD demonstrate significant 685 reduction in the number of boundary hits, especially that of 1.6: 0.6% (6.8%) for Mezaira, 2.5% (0.9%) for GSFC, and 0.4%(4.3%) for Mongu.

The meaning of the sentences is difficult to understand.

Line 715 For this reason, the U27 estimates for the RRI are not reported in the AERONET database.

Is the AERONET database of the U27 estimates publicly available?

Line 816. A climatological LUT was generated from the entire Level 2 AERONET almucantar and hybrid scan database by binning U27s in AOD (440 nm), Angstrom

Exponent (AE, 440-870nm), and SSA (440, 675, 870, 1020 nm).

What is the contribution of hybrid scans to LUT?

Minor comments

Line 280. Figure 3 illustrates the sensitivity of normalized sky radiances (NSR). . .

Possibly, not all readers are familiar with the notion "normalized sky radiances". Please define NSR in the text or present a reference.

Subsection 3.4 Please make the text in the section consistent with table numbering. In the text, the aerosol parameters are considered in the following order: SSA, RRI, parameters of particle size distribution. Tables are presented in a different order: SSA, parameters of particle size distribution, RRI.

Figure 3, Figure 22 Curves, corresponding to the same wavelength, are better to give in the same color.

Throughout the text. Please explain what abbreviations (VMR, etc.) stand for, not in the figure captions, but in the text of the manuscript

This manuscript is very useful and is recommended for publication. However, the authors are hoped to consider our comments or to argue why they are unreasonable to consider.

Please also note the supplement to this comment:
https://www.atmos-meas-tech-discuss.net/amt-2019-474/amt-2019-474-RC1-supplement.pdf

---

## Referee Comment (RC2) · Anonymous Referee #1 · 10 Feb 2020

The manuscript describes the AERONET version 3 aerosol retrieval algorithm. The AERONET network, database and data retrieval constitute a unique source for world wide aerosol information. The version 3 algorithm provides new features and refinements that will increase the value of the data set. Specifically, the new hybrid measurement method is of great interest as it extends the set of solar zenith angles and possible atmospheric conditions for which measurements may be made.

The manuscript is well-organised and include detailed description of the new version aerosol retrieval algorithm and a thorough comparison with version 2 of the algorithm. The manuscript is acceptable for publication with only minor changes as suggested

below.

**Minor comments**

- **Page 6, line 193-194**: Please describe how high performance multi-processor computing is utilized.

- **Page 8, line 251**: what is meant by 'the effect of high range of instrument sensor head temperature'?

- **Page 17, line 544**: Please justify why you make this the only choice as a proxy for the uncertainty.

- **Pages 23-26**: It might be be of interest to mention which data will be reprocessed with the V3 algorithm and if there is a timeplan. Furthermore, are there any plans for further deployment of instruments that may use the HYB protocol?

- **Pages 52-80**: In all Figures with two or more plots there is no a), b) (or c)) labels on the plots. Please include.

- **Pages 56, Fig 6**: Please use same scale on y-axis on a) and b) to make comparison easier.

- **Pages 69-80, Figs 19-21. 23-30**: These plots may provide more information if the points are colored according to the density of points, compare the left and right plots in Fig. 1 below. They show the same data, but information content is higher when plotted as in the left plot.

**Language corrections**

- **Page 4, line 109**: change 'used by aerosol' to 'used by the aerosol'.

- **Page 5, line 148**: change 'and an extend' to 'and extend'.

- **Page 6, line 185**: change 'of electromagnetic' to 'of the electromagnetic'.

- **Page 6, line 192**: change 'to potential' to 'to the potential'.

- **Page 6, line 196**: change 'in UV is possible by adding 380 nm channel to standard'to 'in the UV is possible by adding the 380 nm channel to the standard'.

- **Page 6, line 197**: change 'in UV' to 'in the UV'.

- **Page 6, line 201**: change 'in UV' to 'in the UV'.

- **Page 7, line 217**: change '1°.' to '1.0°'.

- **Page 7, line 239**: change 'to Ross-Li' to 'to the Ross-Li'.

- **Page 8, line 243**: change 'by MODIS' to 'by the MODIS'.

- **Page 8, line 250**: change 'observations the V3' to 'observations V3'.

- **Page 8, line 271**: change 'increasing of wavelength' to 'increasing wavelength'.

- **Page 8, line 272**: change 'of BRDF' to 'of the BRDF'.

- **Page 9, line 275**: change 'of surface' to 'of the surface'.

- **Page 9, line 281**: change 'of SZA' to 'of the SZA'.

- **Page 9, line 282**: change 'when AOD at 440nm is 0.65 and Angstrom' to 'when the AOD at 440nm is 0.65 and the Angstrom'.

- **Page 9, line 283**: change 'increasing of SZA' to 'increasing SZA'.

- **Page 9, line 296**: change 'from the Fig.' to 'from Fig.'.

- **Page 9, line 297**: change 'which the' to 'which is the'.

- **Page 9, line 298**: change 'is a latest' to ''is the latest.

- **Page 10, line 306**: change 'of above' to 'of the above'.

- **Page 10, line 306**: change 'illustrated on a' to 'illustrated by a'.

- **Page 10, line 307**: change 'during UAE' to 'during the UAE'.

- **Page 10, line 311**: change 'of temperature' to 'of the temperature'.

- **Page 10, line 313**: change 'afternoon. On Fig' to 'afternoon. In Fig'.

- **Page 10, line 314**: change 'Figure6a' to 'Figure 6a'.

- **Page 10, line 316**: change 'to temperature' to 'to the temperature'.

- **Page 10, line 339**: change 'by AERONET' to 'by the AERONET'.

- **Page 11, line 347**: change 'with decreasing of the range of the' to 'with decreasing range of the'.

- **Page 11, line 350**: change 'where fit' to 'where the fit'.

- **Page 11, line 353**: change 'shows diurnal' to 'shows the diurnal'.

- **Page 11, line 359**: change 'of total' to 'of the total'.

- **Page 11, line 365**: change 'for Hamim' to 'for the Hamim'.

- **Page 11, line 368**: change 'results shown at Fig' to 'results in Fig'.

- **Page 11, line 371**: change 'shown at Fig. 10a' to 'shown in Fig. 10a'.

- **Page 11, line 372**: change 'shown at Fig. 10b' to 'shown in Fig. 10b'.

- **Page 12, line 390**: change 'refractive index' to 'the refractive index'.

- **Page 12, line 398**: change 'part of refractive' to 'part of the refractive'.

- **Page 12, line 407**: change 'which explained' to 'which is explained'.

- **Page 13, line 409**: change 'make dependence' to 'make the dependence'.

- **Page 13, line 415**: change 'by inversion' to 'by the inversion'.

- **Page 13, line 428**: change 'in retrieved' to 'in the retrieved'.

- **Page 13, line 429**: change 'increasing of wavelength' to 'increasing wavelength'.

- **Page 14, line 442**: change 'is the function of SZA' to 'is a function of the SZA'.

- **Page 15, line 477**: change 'from another hand' to 'on the other hand'.

- **Page 15, lines 479-480**: This sentence may be written as: 'This is illustrated in Fig. 13b which shows that for Mongu Inn, the SSA does not exhibit any significant SZA dependence'.

- **Page 15, line 486**: change 'by changing in' to 'by changes in'.

- **Page 15, line 487**: change 'at 1020 nm, 30° SZA bin' to 'at 1020 nm, the 30° SZA bin'.

- **Page 15, line 498**: change 'decreasing of the difference' to 'decreasing differences'.

- **Page 16, line 516**: change 'participating in' to 'participating in the'.

- **Page 16, line 538**: change 'In Ross-Li' to 'In the Ross-Li'.

- **Page 21, line 673**: change 'uncertainty estimating does' to 'uncertainty estimation does'.

- **Page 21, line 683**: change 'hit 1.6' to 'hit the 1.6'.

- **Page 21, line 698**: change 'though' to 'through'.

- **Page 21, line 704**: change 'in GSFC' to 'in the GSFC'.

- **Page 23, line 745**: change 'Fig.29' to 'Fig. 29'.

- **Page 23, line 749**: change 'Fig.30' to 'Fig. 30'.

- **Page 24, line 777**: change 'At 1020 nm' to 'For the 1020 nm'.

[Figure]

**Fig. 1.** Example of data points colored according to density (left) and not (right).

---

## Referee Comment (RC3) · Anonymous Referee #4 · 15 Feb 2020

The manuscript presents the details of the new AERONET v3 algorithm, comparing the results that can be obtained with those from the previous v2. It also evaluates the uncertainties associated with the parameters that can be obtained by the v3 inversion at different wavelengths, most of them not knows before (v2).

Even recognizing the importance and the accuracy of the present work, the general impression is that most of the results need a deeper discussion, while most if the times the compression is left to the reader. On the other hand, the manuscript is already very long and its extension not desirable. One solution could be to split it into 2 parts, the description on the algorithm and the evaluation of uncertainties.

[Figure]

The length and the amount of material presented in the manuscript is so high that it also makes necessary to better organize the tables and figures. In general, I suggest the following:

* Tables 1.1 to 4.3 could be grouped in some way, by parameter of by site. Same for Table 6.1 to 6.3.

* The same apply for figures 9-10, 13 to 16, 17-18 and 26 to 29.

* In order to facilitate comparisons, all the panels in the same figure should have the same y range when showing the same variable.

* Also, it is desirable to use a color scheme for the different wavelength throughout all the figures.

The caption of the tables 1.1 to 4.3 speaks of statistic, without specify that the shown values are averages.

---

## Referee Comment (RC4) · Anonymous Referee #5 · 18 Feb 2020

The paper is, in the main, well written and a valuable resource for the wide Aeronet user community and the larger atmospheric aerosol community in general.

The other reviewers have done a thorough job reviewing the manuscript. I will only add a few points here.

1) As noted by reviewer #4 that the results need a deeper discussion, however, the paper is already long. Reviewer #4 suggests splitting the paper into two parts, the description on the algorithm and the evaluation of uncertainties.

In general, I see the merit in this suggestion. I would add that the authors should consider the use of appendices for more detailed discussions. Doing this could improve

the readability of the paper without compromising the level of detail for the 'expert' reader.

2) Reviewer 4 also makes points out the shortcomings in the scatter plots i.e. when so many points are plotted all it results in a 'blob' being produced and the information as to the number of observations represented is lost. I strongly agree with Reviewer 4's recommendation regarding this point.

Editorial issues not noted by other referees.
* * *
Line 40. Insert a comma after 'statistics'.

Line 340: 'the backscattering direction.'

Line 450: 'new' ==> 'newly'

Line 456: 'by varying the azimuth angle similar to that of an ALM scan except than the view angle is not equal to SZA.

---

## Author Comment (AC1) · 14 Apr 2020

The AERONET Version 3 aerosol retrieval algorithm, associated uncertainties and comparisons to Version 2" by A. Sinyuk et al.

Authors would like to thank reviewer for careful reading of the manuscript and valuable comments.

General comments. "The topics in the manuscript are undoubtedly urgent. The problem is that certain questions within each of these tasks require a more detailed description than is done in this version of the manuscript. I think it is reasonable to

submit a number of considered questions either after, or in parallel with, publishing this manuscript. It is hardly appropriate to expand substantially the given version of the manuscript. In particular, the manuscript already contains very many figures and tables, complicating the perception of the material. The second variant is to divide the text presented into two parts: description of the AERONET Version 3 and description of new approach to estimating the uncertainties in the retrieved aerosol parameters. In principle, these are two different tasks that can be described separately."

Answer:

The main objective of this paper is to present a description of V3 AERONET aerosol retrieval algorithm including all the changes and new additions. In this respect, the estimation of retrieval uncertainties is a part of the V3 aerosol retrieval algorithm and should be a part of this manuscript rather than a separate publication. We understand that combining all the parts of V3 aerosol retrieval algorithm in one manuscript does not allow for discussion of every detail and nuance. However, we did our best to provide a reasonable number of details in describing each part of the algorithm. It might well happen that during further research some more details of uncertainty estimates and other parts of the V3 algorithms may be included in future publications. At this point, however, we believe that separation of the manuscript in two parts is not appropriate.

Major comments.

1. Line 223. While for the almucantar (ALM) observation geometry this is a reasonable assumption (e.g. Dubovik and King, 2000b; Torres et al., 2014), for other geometries the sensitivity to vertical structure of aerosol and gases in atmosphere can be impor-tant, especially at shorter wavelengths with relatively large Rayleigh scattering. Please provide a reference or numerical estimates

Answer:

The following sentence was added:

For example, the effect of aerosol vertical distribution on aerosol parameters retrieved from ALM and principle plane (PP) observations was analyzed in (Torres et al., 2014). It was shown that PP retrievals are more strongly affected by assumptions on aerosol vertical distribution than those of ALM.

2. Page 9 (3.2 Effects of changes in Extraterrestrial solar flux and temperature correction) Figure 5 discusses the temperature dependences of NSR. Seemingly, the temperatures in the range of 10-50o C at the selected site are distributed non-uniformly, and points on the plots differ in statistical representativeness. Can this influence the result?

Answer:

In generating Figure 5 no statistic was used. Instead: "Each point corresponds to an individual observation taken at specific value of the sun photometer sensor head temperature.". This sentence was added to discussion of Figure 5. As for the sampling, all temperature trends are smooth and consistent, and if there was a sampling problem then the data would look noisy.

3. Section 4 (Hybrid scan: concept and retrieval scan). From materials, now presented in section 4, we can gain only a general idea of the novelties associated with the new instrument type. If they are left in about the same form that we see in the manuscript, it is reasonable to consider questions, regarding the hybrid scan, in more detail and, presumably, in a separate publication. Maybe this publication does already exist now?

Answer:

A separate publication on HYB scan does not currently exists. However, we think a future HYB publication is possible especially when more statistics from T model sun-photometers will become available.

4. How many photometers, ensuring the new scanning geometry, have been installed and already operate now? Are they installed at all four sites, data from which are used

in the present study, and how long ago? Does the aeronet.gsfc.nasa.gov site present information on the type of instrument (or instruments, if they operate in parallel) that is used at arbitrarily chosen observation site? In any case, I cannot see such data. Did the cloud screening procedure change after passing to new sensing geometry?

Answer:

Information on AERONET sites with T-model sun-photometers installed and information on AERONET website, the following sentence was added:

"The information on the AERONET sites equipped with Model-T sun photometers can be found on the AERONET web site which provides an option to choose between ALM and HYB scan scenarios (https://aeronet.gsfc.nasa.gov/cgi-bin/draw_map_display_inv_v3 ). "

Information on four AERONET sites used in analysis, the following sentence was added: These sites have the longest record of HYB type observations starting from the fall of 2014.

Information on T-model instruments, the following sentence was added:

Descriptions of this model is provided on the AERONET website: https://aeronet.gsfc.nasa.gov/new_web/system_descriptions_instrument.html.

Cloud screening procedures are the same for both the ALM and HYB scan geometries.

5. Results in subsection 4.2 are described very sparingly. Results in Figs. 17 and 18 can hardly be considered as an argument in favor of the good correspondence between results retrieved using two different scanning geometries, more so considering that the authors present no statistics that was used in these comparisons. For instance, it is unclear why the authors conclude that ÃŘĔĞrt the same time the variability is increasing with increasing wavelength due to predominant contribution of fine mode aerosols to the generated statistics, and therefore much smaller AOD at the longest wavelengths which results in less sensitivity to aerosol absorption (line 496)?

Answer:

The following sentences were modified:

The statistics were generated using the data from all AERONET sites for which the HYB inversions were available by aggregating SSA retrievals in five SZA bins. Each bin is 10 wide centered at 50.5° (387), 54.5° (160), 59.5° (187), 64.5° (121) and 73.5° (296) SZA where the number of inversions corresponding to each SZA bin are shown in parentheses.

At the same time the variability is increasing with increasing wavelength due to predominant contribution of fine mode aerosols to the generated statistics (the aerosol loading of ∼80% of sites is dominated by fine mode aerosols), and therefore much smaller AOD at the longest wavelengths which results in less sensitivity to aerosol absorption.

6. Can you present data for at least a few observation sites that would show how much the number of retrievals increased after the new geometry was introduced?

Answer:

The following sentence was added:

The extension of the SZA range in HYB scan geometry results in substantial increase in the number of inversions: e. g. Mezaira (∼61%) and Kanpur (∼ 57%). The increase was estimated as the ratio of the number of inversions for SZA less than 50° to that for larger SZAs.

7. Line 534. The radiometric calibration and solar spectrum irradiance uncertainties are combined in one bias because both of them affect the magnitude of sky radiances. What is the value of uncertainty, resulting from combining the radiometric calibration and solar spectrum irradiance uncertainties?

Answer:

We did not estimate the contribution to uncertainties from different sources of biases

separately. Therefore, we added a qualitative statement in discussion as follows:

The combined calibration and solar irradiance bias assume the ± 5% values, which overestimate the sum of individual biases in cases when they compensate each other.

8. Subsection 5.4. In analyzing the results from retrieving aerosol characteristics, it is important for the reader to obtain information on what is the uncertainty degree of any characteristic at a specific site under certain atmospheric conditions. The authors carried out such an analysis for four sites (GSFC, Mezaira, Kanpur, and Mongu), which correspond to four different aerosol types. It is reasonable to stress this in the text of the manuscript (put simply, starting an analysis, say, for GSFC, to prescribe aerosol type characteristic for this observation site in an explicit form). It would also be useful for the reader to see any climatologic data on characteristics for four selected sites (type of AOD distribution, relationship between fine and coarse fractions, characteristic types of the underlying surface, etc.) Then it will be easier for the reader to choose estimates of uncertainty that correspond to the specific data that he analyzes.

Answer:

We added a new table which summarizes the multi-year averages of aerosol and surface characteristics at four AERONET sites used in analysis:

The analysis is supplemented by Table 18 which contains multiyear (1995-2019) averages of aerosol and surface characteristics at four selected AERONET sites. The following abbreviations are used in the Table 14 header: FMF stands for the fine mode fraction of AOD, AE is Angstrom exponent estimated for (440-879) wavelength range, and SA is a surface albedo.

Table 18. Multiyear (1995-2019) averages of aerosol and surface characteristics at four AERONET sites selected for analysis of uncertainties in retrieved aerosol parameters. The abbreviations FMF, AE and SA stand for the fine mode fraction of AOD, Angstrom exponent estimated for (440-879) wavelength range, and surface albedo respectively.

9. Line 666. Fig. 24 shows the uncertainties (by the U27 methodology) of the RRI at 440 nm estimated for the GSFC, Mongu, and Mezaira sites. Can any recommendations be obtained regarding RRI at other wavelengths?

Answer:

The following discussion was added:

The U27 estimated for RRI at longer wavelengths are very similar to those at 440 nm. The similarity between estimated uncertainties for RRI at different wavelengths can be explained by the fact that spectrally RRI retrievals are not independent but related through the smoothness constrains (e. g. Dubovik and King, 2000).

10. Line 679. In contrast, the U27 estimated for SSA and observed variability of SSA retrievals are more consistent. Additionally, the upper and lower limit retrieval constraints of RRI (1.33 and 1.60, respectively) were reached at times, suggesting some potential instability in the retrievals of RRI. The percentage values of RRI retrievals reaching 1.33 and 1.6 (in parentheses) were estimated for all three sites for 440 nm AOD less than 0.2 and greater than 0.4. The estimates for smaller AOD bin show that RRI retrievals hit 1.6 boundary more often than that of 1.33: 1.6% (13.7%) for Mezaira, 3.1% (11.1%) for GSFC, and 2.2% (18.5%) for Mongu. The corresponding estimates for larger AOD demonstrate significant 685 reduction in the number of boundary hits, especially that of 1.6: 0.6% (6.8%) for Mezaira, 2.5% (0.9%) for GSFC, and 0.4% (4.3%) for Mongu. The meaning of the sentences is difficult to understand.

Answer:

The paragraph was modified as follows:

The potential instability in the retrievals of RRI is further illustrated by the fact that the upper and lower limits imposed on RRI variability (1.33 and 1.60, respectively) are often reached especially for low AOD. The percentage values of RRI retrievals reaching 1.33 and 1.6 (in parentheses) were estimated for all three sites for two bins in 440 nm

AOD: less than 0.2 and greater than 0.4. The estimates for smaller AOD bin show that RRI retrievals hit the 1.6 boundary more often than that of 1.33: 1.6% (13.7%) for Mezaira, 3.1% (11.1%) for GSFC, and 2.2% (18.5%) for Mongu. The corresponding estimates for larger AOD demonstrate significant reduction in the number of boundary hits, especially that of 1.6: 0.6% (6.8%) for Mezaira, 2.5% (0.9%) for GSFC, and 0.4% (4.3%) for Mongu.

11. Line 715 For this reason, the U27 estimates for the RRI are not reported in the AERONET database. Is the AERONET database of the U27 estimates publicly available?

Answer:

The following sentence was added to the conclusion part:

The Level 2 U27 estimates for retrieval uncertainty are available at AERONET website (https://aeronet.gsfc.nasa.gov/cgi-bin/webtool_inv_v3) for each AERONET site.

12. Line 816. A climatological LUT was generated from the entire Level 2 AERONET almucantar and hybrid scan database by binning U27s in AOD (440 nm), Angstrom Exponent (AE, 440-870nm), and SSA (440, 675, 870, 1020 nm). What is the contribution of hybrid scans to LUT?

Answer:

The following sentence was modified:

A climatological LUT was generated from the entire Level 2 AERONET almucantar and hybrid ($\sim$ 10% of total scans) scan database by binning U27s in AOD (440 nm), Angstrom Exponent (AE, 440- 870nm), and SSA (440, 675, 870, 1020 nm).

Minor comments

1. Line 280. Figure 3 illustrates the sensitivity of normalized sky radiances (NSR): :Possibly, not all readers are familiar with the notion "normalized sky radiances". Please

define NSR in the text or present a reference.

Answer:

The following sentence was modified to include NSR definition:

Figure 3 illustrates the sensitivity of normalized sky radiances (NSR), to the changes in BRDF parameters. The NSR constitute the input to the inversion code and are defined as the measurements divided by extraterrestrial solar flux and multiplied by $\pi$.

2. Subsection 3.4 Please make the text in the section consistent with table numbering. In the text, the aerosol parameters are considered in the following order: SSA, RRI, parameters of particle size distribution. Tables are presented in a different order: SSA, parameters of particle size distribution, RRI.

Answer:

The order of tables was changed.

3. Figure 3, Figure 22 Curves, corresponding to the same wavelength, are better to give in the same color.

Answer:

The colors were changed.

4. Throughout the text. Please explain what abbreviations (VMR, etc.) stand for, not in the figure captions, but in the text of the manuscript.

Answer:

Definitions were included in the text.

Please also note the supplement to this comment:
https://www.atmos-meas-tech-discuss.net/amt-2019-474/amt-2019-474-AC1-supplement.pdf

---

## Author Comment (AC2) · 14 Apr 2020

The AERONET Version 3 aerosol retrieval algorithm, associated uncertainties and comparisons to Version 2" by A. Sinyuk et al.

Reply to reviewer #1.

Authors would like to thank reviewer for careful reading of the manuscript and valuable comments.

1. Page 6, line 193-194: Please describe how high-performance multi-processor computing is utilized.

[Figure]

The sentence describing high performance computing was replaced by the following:

" The NASA Center for Climate Simulation (NCCS) at the NASA Goddard Space Flight Center (GSFC) Discover cluster is utilized to overcome the significant increase in computational resources required for the application of SORD in AERONET retrievals (https://www.nccs.nasa.gov/systems/discover)."

2. Page 8, line 251: what is meant by 'the effect of high range of instrument sensor head temperature'?

We modified the corresponding sentence as follows:

"In order to improve the overall quality of observations V3 employs a temperature correction of both AOD and sky measurements applied to account for the temperature sensitivity of detectors and filters of instruments under conditions of high sensor head temperature variability $\sim$ (-25 to +55°C) in various environments. "

3. Page 17, line 544: Please justify why you make this the only choice as a proxy for the uncertainty.

We added the following explanation to the corresponding sentence:

"From these statistics only the standard deviation (which we label as U27) is used as a proxy for the estimated uncertainty as an indicator of a spread of the retrievals corresponding to the different combinations of input uncertainties."

4. Pages 23-26: It might be be of interest to mention which data will be reprocessed with the V3 algorithm and if there is a timeplan. Furthermore, are there any plans for further deployment of instruments that may use the HYB protocol?

We added the following paragraph at the end of the manuscript addressing these and some other questions.

"As of now the entire AERONET data base is reprocessed using V3 aerosol retrieval algorithm. The Level 2 U27 estimated are available at AERONET website

(https://aeronet.gsfc.nasa.gov/cgi-bin/webtool_inv_v3) for each AERONET site. The number of new CIMEL Model-T instruments that can take measurements in HYB protocol is steadily increasing. "

5. Pages 52-80: In all Figures with two or more plots there is no a), b) (or c)) labels on the plots. Please include.

Included.

6. Pages 56, Fig 6: Please use same scale on y-axis on a) and b) to make comparison easier.

Corrected.

7. Pages 69-80, Figs 19-21. 23-30: These plots may provide more information if the points are colored according to the density of points, compare the left and right plots in Fig. 1 below. They show the same data, but information content is higher when plotted as in the left plot.

All the plots with high density of points were modified to add colored density, see example below:

(see supplement)

8. Language corrections.

All suggested language corrections are incorporated.

Please also note the supplement to this comment:
https://www.atmos-meas-tech-discuss.net/amt-2019-474/amt-2019-474-AC2-supplement.pdf
* * *

---

## Author Comment (AC3) · 14 Apr 2020

The AERONET Version 3 aerosol retrieval algorithm, associated uncertainties and comparisons to Version 2" by A. Sinyuk et al.

Reply to reviewer #4.

The authors would like to thank the reviewer for his/her diligent reading of the manuscript and useful comments.

Comments: 1. Even recognizing the importance and the accuracy of the present work, the general impression is that most of the results need a deeper discussion, while most

if the times the compression is left to the reader. On the other hand, the manuscript is already very long and its extension not desirable. One solution could be to split it into 2 parts, the description on the algorithm and the evaluation of uncertainties.

Answer:

The main objective of this paper is to present a description of V3 AERONET aerosol retrieval algorithm including all the changes and new additions. In this respect, the estimation of retrieval uncertainties is a part of the V3 aerosol retrieval algorithm and should be a part of this manuscript rather than a separate publication. We understand that combining all the parts of V3 aerosol retrieval algorithm in one manuscript does not allow for discussion of every detail and nuance. However, we did our best to provide a reasonable number of details in describing each part of the algorithm. It might well happen that during further research some more details of uncertainty estimates and other parts of the V3 algorithms may be included in future publications. At this point, however, we believe that separation of the manuscript in two parts is not appropriate.

2. The length and the amount of material presented in the manuscript is so high that it also makes necessary to better organize the tables and figures. In general, I suggest the following: * Tables 1.1 to 4.3 could be grouped in some way, by parameter of by site. Same for Table 6.1 to 6.3. * The same apply for figures 9-10, 13 to 16, 17-18 and 26 to 29.

Answer:

The suggestions in these comments are very general which makes it unclear what specific changes are meant. In particular, the tables and figures are already grouped by parameters and sites and, as we believe, give a rather clear and detailed illustration to the discussions in the text of the manuscript.

3. In order to facilitate comparisons, all the panels in the same figure should have the same y range when showing the same variable.

Answer:

The corresponding plots were modified.

4. Also, it is desirable to use a color scheme for the different wavelength throughout all the figures.

Answer:

The same color scheme is now used for different wavelengths.

5. The caption of the tables 1.1 to 4.3 speaks of statistic, without specify that the shown values are averages.

Answer:

The captions to corresponding tables were corrected, for example:

Table 6. Statistics, average values and standard deviations (in parentheses), of the difference in volume median radius (VMR) and width of particle size distribution (STD) retrievals of V2 and V3 for GSFC site. The difference is defined as V3 -V2. The numbers in parentheses are standard deviations.

Please also note the supplement to this comment:
https://www.atmos-meas-tech-discuss.net/amt-2019-474/amt-2019-474-AC3-supplement.pdf

---

## Author Comment (AC5) · 14 Apr 2020

The AERONET Version 3 aerosol retrieval algorithm, associated uncertainties and comparisons to Version 2" by A. Sinyuk et al.

Reply to reviewer #5.

The authors would like to thank the reviewer for carefully reading the manuscript and valuable comments.

Comments:

1. As noted by reviewer #4 that the results need a deeper discussion, however, the

paper is already long. Reviewer #4 suggests splitting the paper into two parts, the description on the algorithm and the evaluation of uncertainties. In general, I see the merit in this suggestion. I would add that the authors should consider the use of appendices for more detailed discussions. Doing this could improve the readability of the paper without compromising the level of detail for the 'expert' reader.

Answer:

The main objective of this paper is to present a description of V3 AERONET aerosol retrieval algorithm including all the changes and new additions. In this respect, the estimation of retrieval uncertainties is a part of the V3 aerosol retrieval algorithm and should be a part of this manuscript rather than a separate publication. We understand that combining all the parts of V3 aerosol retrieval algorithm in one manuscript does not allow for discussion of every detail and nuance. However, we did our best to provide a reasonable number of details in describing each part of the algorithm. It might well happen that during further research some more details of uncertainty estimates and other parts of the V3 algorithms may be included in future publications. At this point, however, we believe that separation of the manuscript in two parts is not appropriate.

2. Reviewer 4 also makes points out the shortcomings in the scatter plots i.e. when so many points are plotted all it results in a 'blob' being produced and the information as to the number of observations represented is lost. I strongly agree with Reviewer 4's recommendation regarding this point.

Answer:

All the scatter plots with large number of points were replotted according to the recommendation, see example below.

(see supplement

Please also note the supplement to this comment:

https://www.atmos-meas-tech-discuss.net/amt-2019-474/amt-2019-474-AC5-supplement.pdf

---

## Author Comment (AC4)

The AERONET Version 3 aerosol retrieval algorithm, associated uncertainties and comparisons to Version 2" by A. Sinyuk et al.

Reply to reviewer #1.

Authors would like to thank reviewer for careful reading of the manuscript and valuable comments.

1. **Page 6, line 193-194**: Please describe how high-performance multi-processor computing is utilized.

   The sentence describing high performance computing was replaced by the following:

" The NASA Center for Climate Simulation (NCCS) at the NASA Goddard Space Flight Center (GSFC) Discover cluster is utilized to overcome the significant increase in computational resources required for the application of SORD in AERONET retrievals (https://www.nccs.nasa.gov/systems/discover)."

2. **Page 8, line 251**: what is meant by 'the effect of high range of instrument sensor head temperature'?

   We modified the corresponding sentence as follows:

   "In order to improve the overall quality of observations V3 employs a temperature correction of both AOD and sky measurements applied to account for the temperature sensitivity of detectors and filters of instruments under conditions of high sensor head temperature variability ~ (-25 to +55°C) in various environments. "

3. **Page 17, line 544**: Please justify why you make this the only choice as a proxy for the uncertainty.

   We added the following explanation to the corresponding sentence:

 "From these statistics only the standard deviation (which we label as U27) is used as a proxy for the estimated uncertainty as an indicator of a spread of the retrievals corresponding to the different combinations of input uncertainties."

4. **Pages 23-26**: It might be be of interest to mention which data will be reprocessed with the V3 algorithm and if there is a timeplan. Furthermore, are there any plans for further deployment of instruments that may use the HYB protocol?

   We added the following paragraph at the end of the manuscript addressing these and some other questions.

"As of now the entire AERONET data base is reprocessed using V3 aerosol retrieval algorithm. The Level 2 U27 estimated are available at AERONET website (https://aeronet.gsfc.nasa.gov/cgi-bin/webtool_inv_v3) for each AERONET site. The number of new CIMEL Model-T instruments that can take measurements in HYB protocol is steadily increasing. "

5. **Pages 52-80**: In all Figures with two or more plots there is no a), b) (or c)) labels on the plots. Please include.

Included.

6.  **Pages 56, Fig 6**: Please use same scale on y-axis on a) and b) to make comparison easier.

Corrected.

7.  **Pages 69-80, Figs 19-21. 23-30**: These plots may provide more information if the points are colored according to the density of points, compare the left and right plots in Fig. 1 below. They show the same data, but information content is higher when plotted as in the left plot.

All the plots with high density of points were modified to add colored density, see example below:

[Figure]

Figure 30. Uncertainties for a) volume median radius of coarse mode and b) width of size distribution for coarse mode estimated at Kanpur site.

8.  **Language corrections**.

All suggested language corrections are incorporated.